# DNA Double-Strand Breaks as Pathogenic Lesions in Neurological Disorders

**DOI:** 10.3390/ijms23094653

**Published:** 2022-04-22

**Authors:** Vincent E. Provasek, Joy Mitra, Vikas H. Malojirao, Muralidhar L. Hegde

**Affiliations:** 1Department of Neurosurgery, Center for Neuroregeneration, Houston Methodist Research Institute, Houston, TX 77030, USA; vprovasek@houstonmethodist.org (V.E.P.); vhhmalojirao@houstonmethodist.org (V.H.M.); 2College of Medicine, Texas A&M University, College Station, TX 77843, USA; 3Department of Neurosciences, Weill Cornell Medical College, New York, NY 11021, USA

**Keywords:** TDP-43, hnRNPs, DNA double-strand break repair, DNA damage response, neurodegeneration, dementia

## Abstract

The damage and repair of DNA is a continuous process required to maintain genomic integrity. DNA double-strand breaks (DSBs) are the most lethal type of DNA damage and require timely repair by dedicated machinery. DSB repair is uniquely important to nondividing, post-mitotic cells of the central nervous system (CNS). These long-lived cells must rely on the intact genome for a lifetime while maintaining high metabolic activity. When these mechanisms fail, the loss of certain neuronal populations upset delicate neural networks required for higher cognition and disrupt vital motor functions. Mammalian cells engage with several different strategies to recognize and repair chromosomal DSBs based on the cellular context and cell cycle phase, including homologous recombination (HR)/homology-directed repair (HDR), microhomology-mediated end-joining (MMEJ), and the classic non-homologous end-joining (NHEJ). In addition to these repair pathways, a growing body of evidence has emphasized the importance of DNA damage response (DDR) signaling, and the involvement of heterogeneous nuclear ribonucleoprotein (hnRNP) family proteins in the repair of neuronal DSBs, many of which are linked to age-associated neurological disorders. In this review, we describe contemporary research characterizing the mechanistic roles of these non-canonical proteins in neuronal DSB repair, as well as their contributions to the etiopathogenesis of selected common neurological diseases.

## 1. Introduction: DNA Double-Strand Break (DSB) Repair in the Central Nervous System (CNS)

Genomic stability is crucial for the maintenance of homeostasis and normal physiological functions of cells and tissues throughout the body. While the study of DNA damage has been classically associated with neoplastic pathologies, its role in neurological disease has been increasingly appreciated by a growing body of literature. Studies characterizing the functions of DNA repair processes in nondividing, post-mitotic neurons have unlocked a new understanding of neuronal biology and associated pathologies that include Alzheimer’s disease (AD), Parkinson’s disease (PD), amyotrophic lateral sclerosis (ALS), and brain aging. In this section, we will describe how neurons in basal and pathological states respond to breaks in both strands of the DNA molecule, and how these breaks affect brain physiology. Furthermore, we will highlight the roles of key proteins involved in the recognition, repair, and signaling of neuronal DSBs, as well as discuss how these processes are affected by chromatin dynamics.

### 1.1. Functions and Resolution of DSBs in Neurons

Cells experience a near-constant assault on the integrity of the genome, resulting in the accumulation of broken DNA strands that interfere with cellular functions. DSBs are the most lethal form of DNA damage, and mammalian cells are estimated to accumulate up to 50 DSBs per day [1]. If left unrepaired, these aberrant DNA structures always lead to cell cycle arrest and apoptosis.

DSBs can occur either directly by the action of a DSB inducing agent (e.g., the topoisomerase II inhibitor, etoposide) or indirectly by the conversion of single-strand breaks (SSBs) into a DSB lesion [2]. Whereas direct DSBs typically initiate a defined set of damage-specific repair pathways, indirect DSBs require activation and integration of multiple pathways to resolve both SSBs and DSBs simultaneously, particularly when clustered with other oxidative damage [3]. Direct DSBs are repaired via two canonical pathways: homologous recombination (HR) or non-homologous end-joining (NHEJ). The choice of the repair pathway largely depends on the cell cycle status. The HR pathway occurs in dividing cells, where it is restricted to the late S and G2 phases of the cell cycle. Of the canonical repair pathways, the HR product exhibits the greatest fidelity to the original DNA sequence. This error-free result is achieved primarily through the use of homologous sequences found in sister chromatids that flank the break site during cell replication [4] as well as in the RAD52-mediated and RNA-templated repair process [5,6]. 

In the CNS, HR has been linked to both disease progression of glioblastoma multiforme (GBM) but also critical to the cells of the developing and aging brain. In GBM, chemotherapeutic targeting of the HR pathway has become a mainstay of treatment, and upregulation of HR factors tends to correlate with worse prognosis and resistance to DNA alkylating therapies. Interestingly, targeting other DNA repair pathways, such as the NHEJ pathway, has no proven therapeutic benefit [7,8,9,10]. On the other hand, in the healthy developing brain, as well as the aging brain, HR is key to maintaining a genetically healthy neuronal stem and progenitor cell populations that are vital for continued learning, memory, and other cognitive functions. While HR is the primary mechanism of DNA repair in these early cell types, the post-mitotic neurons into which they develop primarily rely on the error-prone NHEJ pathway, and its associated factors, such as Ku 70/80 heterodimer, XRCC4, and DNA Ligase 4. Unlike HR, the NHEJ pathway remains active throughout the cell cycle, including in the early S and G1 phases [11]. However, NHEJ is considered less reliable, because its repair product may contain insertion-deletion mutations or even chromosomal translocations [12]. Importantly, while NHEJ is the prevailing mechanism of DSB repair in neurons, a recent study demonstrated that virus-mediated delivery of the CRISPR-Cas9 genome editing system was able to successfully perform homology-directed repair in post-mitotic neurons. Similarly, others have demonstrated the transcription-dependent recruitment of recombination repair factors to oxidative DNA lesions in the neuronal genome, suggesting that an RNA templated HR repair mechanism exists outside of actively replicating cells [5,6]. These findings suggest that neurons possess a repertoire of common DSB repair factors controlled by incompletely characterized regulatory pathways [13]. In the event a neural cell is unable to rely on either of the canonical DSB repair pathways, a separate, non-canonical DSB repair pathway, the microhomology-mediated end-joining (MMEJ), also termed alternative end-joining (AltEJ), generally proceeds, and typically utilizes SSB repair factors to repair DSBs [14,15]. In the absence of NHEJ, particularly in the absence of the Ku70/80 heterodimer, exonuclease-mediated degradation of the broken DNA ends results in single-stranded DNA (ssDNA) overhangs. AltEJ factors, including PARP1 and DNA Polθ, utilize sequences of microhomology between these overhangs to direct ligation of the DNA molecule. This process generally provides a repair product with the least fidelity to the original DNA sequence, and classically leads to microhomology-mediated chromosomal rearrangements. Because of this error-prone nature, the physiological role of AltEJ is typically restricted to immunoglobulin class switching recombination [16,17,18,19]. In neurons, AltEJ-mediated DSB end-joining can be severe enough to induce neuronal apoptosis, and multiple reports have connected the development of AltEJ-related chromosomal translocations to neuropathological abnormalities, including developmental delay, schizophrenia, and affective disorders [20,21]. Despite the severe functional consequences of low fidelity DSB repair, little is known regarding the pathways and conditions that may lead to hyper-activated AltEJ in neurons. 

The functional effects of DSBs are wide and often deleterious. Traditionally, it was thought that breaks in the DNA would always halt cell division and transcription, particularly at sequences adjacent to breaks [22]. New evidence now challenges this notion and posits the counterintuitive point that localized DNA strand breaks and their repair are required for gene activation in certain contexts [23]. Experiments conducted by Bunch et al. demonstrated that DSB-induced signaling pathways were required for the expression of stimulus-inducible genes in humans [24]. Specifically, the enrichment of DSB-associated proteins, γH2AX and phospho-TRIM28, at serum-induced genes was modulated by key DNA repair enzymes: DNA-dependent protein kinase (DNA-PK) and ataxia telangiectasia mutated (ATM). The resulting complex was demonstrated to orchestrate RNA polymerase II (RNAPII) pause-release with subsequent transcriptional elongation of actively transcribed genes [24]. This transcriptional activation-coupled DNA damage response (DDR) signaling suggests that actively transcribed genes require special support in maintaining genetic integrity [25]. Furthermore, Hegde and colleagues have previously demonstrated that ligand-induced gene activation resulted in oxidative damage-mediated SSBs and indirect DSBs via demethylation of histones and cytosine-phosphate-guanines (CpGs) surrounding transcription start sites [23]. The repair of the resulting damage was required for successful gene activation and was preferentially performed in promoter regions. Conversely, the activation of ligand-independent heat shock proteins is associated with the generation of direct DSBs in the promoter region, where efficient repair is likewise required for successful gene expression [23]. Subsequent studies have confirmed those findings, and a genome-wide association study (GWAS) conducted by Wu et al. reported that SSB susceptible DNA sequences were more frequently located within enhancer sequences or near CpG islands of promoter regions in the neuronal genome [26]. These observations are particularly significant for understanding neurogenesis and neural plasticity during development. These processes are regulated by retinoic acid (RA) signaling, where RA receptors and peroxisome proliferator-activated receptor β/δ (PPAR β/δ) are activated in a ligand-inducible manner [27,28]. Furthermore, other studies have identified recurrent DSB clusters within genes involved in synaptic function and cell-cell adhesions that govern spatial and functional behaviors of neurons in developing brains [29]. 

The association of DNA strand breaks and gene expression also extends to post-mitotic neurons of the adult brain. In one study, the formation of DSBs in the promoter region of a subset of immediate early-response genes (IERGs), including *FOS*, *EGR1*, and *NPAS4*, was demonstrated to affect experience-driven synaptic modulations required for successful cognitive functions [30]. Notably, the link between IERG expression and DSBs appears unique to the neuron, where topoisomerase IIβ strictly regulates the induction of DSB formation within topological domains characterized by CCCTC-binding factors (CTCF) before IERG expression is observed. Conversely, the same group demonstrated a serum-induced upregulation of IERGs involved topoisomerase IIα and occurred independently of DSB induction by facilitating the promoter-proximal stalling of RNAPII in retinal pigment epithelial cells [31]. These observations are relevant because the functional effects indirectly mediated by DSB formation are extensive; most IERGs encode transcription factors, including *AP1*, *c-Jun*, *c-Fos*, and *c-Myc*, that regulate gene activation of down-stream, late-response genes (LRGs) (e.g., *FGF1*, *HOMER1*, and *BDNF*) in neurons following environmental stimuli [32,33]. The activation of LRGs, unlike IERGs, is associated with activity-induced oxidative damage at promoter regions, and is crucial for long-lasting phenotypic changes that include: modulating synaptic plasticity, neurite outgrowth, neural circuitry, and balanced excitatory-inhibitory synaptic activities [34]. Taken together, these findings suggest that neurons have evolved multiple mechanisms that utilize DSBs to specifically regulate the expression of the IERG and LRG genes. When this process is considered in the context of the DNA break-induced chromatin opening, it seems reasonable that neurons may utilize DSB-induced acute chromatin openings for immediate genes, whereas oxidative stress and/or SSBs induce slow, partial chromatin openings for the activation of long-term genes.

In non-dividing cells, DSBs participate in the homeostasis of two non-canonical DNA secondary structures that include R-loops and guanine quadruplexes. R-loops are transient 3-strand RNA:DNA hybrid structures that form under physiological conditions when a nascent RNA transcript remains hybridized to the template DNA strand. Importantly, part of this structure includes the creation of a displaced region of non-template single-stranded DNA (ssDNA), which affects genome instability and R-loop resolution. Functionally, R-loop structures are closely associated with gene regulation via chromatin dynamics, transcription factor recruitment, and regulation of RNAPII processing, among others [35]. R-loops are usually processed by RNA and DNA binding proteins (discussed in detail in later sections), many of which are associated with post-mitotic/motor neuron diseases [36]. If the R-loop remains unresolved, the persistent ssDNA flank can initiate the DDR, resulting in a cytidine to uracil transversion. This transversion triggers the formation of a DNA nick, which causes the collapse of the replication fork and possible conversion into a secondary DSB via mismatch repair [37,38]. As a result, R-loops represent a potential source of DSBs that may significantly contribute to genomic instability in non-replicating, transcriptionally active cells. Notably, most R-loops are observed at non-template DNA regions enriched in G over C nucleotides [37,38]. Within these G-rich regions, unique secondary structures comprised of four guanine nucleotides organized in four interspaced tandem repeats may develop. These stable single-stranded secondary structures are termed G-quadruplex or G4 structures. Similar to R-loops, these structures are closely associated with gene regulation, especially when located at promoter sequences of transcriptionally active genes [39]. Some investigations have even suggested that a positive feedback relationship may exist between G4 and R-loop structures. In this scenario, the R-loop-induced exposure of ssDNA regions permits the spontaneous formation of G4 structures that in turn confer resistance to R-loop resolution [39]. Indeed, multiple studies have reported the increased presence of DNA damage and enhanced cytotoxicity in cell lines treated with G4-stabilizing ligands, particularly when the HR response is impaired [40]. These results imply that the R-loop/G4 structure formation may act as a double-edged sword. On one hand, the G4 structure can stabilize the transcription initiation bubble within an R-loop. On the other hand, it may activate the DDR or exacerbate the R-loop-mediated replication stress, ultimately causing DSB formation and synthetic lethality [40,41,42,43]. G4 structures are not exclusively related to genome instability; however, recent studies have characterized their other functions in the context of AD. Specifically, it was demonstrated that G4 sequences are enriched in neurons with reduced polycomb group protein BMI1 expression and/or relaxed chromatin, such as those found in sporadic AD. The G4 complexes were concentrated at transcriptionally active sequences and co-localized with RNAPII. Furthermore, the intergenic G4 structures altered splicing events of the transcribed regions and were associated with decreased neuronal gene expression [44]. Taken together, these studies suggest DSBs and the DDR are closely linked to G4 structures that in turn may contribute to the etiopathogenesis of neurological disease.

### 1.2. DNA Damage Response (DDR) in the Chromatin Context

Over the years, a growing body of evidence has demonstrated how DSB repair takes place as a highly orchestrated balance between dynamic nucleosome organization and DNA damage sensing/repair. This is particularly important in the mammalian genome, as it contains a wide array of evolutionarily conserved specialized chromatin structures, that include actively transcribed genomic regions, replication forks, intergenic regions, telomeres, and highly compact heterochromatin [41]. Following DSB induction, two key signaling pathways are activated to begin the repair process that involve the DDR and cell cycle checkpoint regulation. Within 0–5 min of DSB formation, the MRN complex (MRE11, RAD50, and NBS1) binds to DSB sites and facilitates the recruitment and activation of ATM kinase [42,43]. In cycling cells, human single-strand binding protein 1 (hSSB1) plays a crucial role in stimulating the recruitment of the MRN complex at DSBs by directly binding to NBS1 and modulating the endo-nuclease activity of MRN [44]. In addition, the phosphorylation of Rad17 at Threonine 622 by ATM initiates its direct interaction with the MRN complex via NBS1, thereby enhancing the early recruitment of MRN and amplification of ATM signaling [45]. In the early phase of chromatin remodeling, Bloom Syndrome protein (BLM) is recruited in an MRN/ATM-dependent manner. RNF8-mediated polyubiquitylation of BLM is critical for its helicase activity and interaction with NBS1. Activated BLM plays crucial roles in the DSB repair pathway choice in a cell cycle-stage dependent manner, such as inhibiting the recruitment of HR factors at the S phase and NHEJ factors at the G1 phase [46]. BLM also plays a key role as an apoptosis sensor during DSB induction in post-mitotic, immature cortical neurons [47]. BLM expression is upregulated by the p53 and AP1 signaling pathways during DDR. Moreover, its deficiency leads to the accumulation of oxidative genome damage and mitochondrial fragmentation [48]. Next, activated ATM sequentially phosphorylates a series of signaling proteins involved in the cell cycle checkpoint (e.g., p53 and Chk2) and DDR (e.g., BRCA1 and p53BP1) pathways [49]. Phosphorylation of histone γH2AX at serine 139 (as γH2AX) by ATM kinase results in a binding site for the BRCA1 C-terminal domain of the Mediator of DNA damage Checkpoint protein 1 (MDC1) and its subsequent recruitment to break sites [50,51]. The positioning of MDC1 is critical for the efficient recruitment of the MRN complex and ATM kinase [52,53]. Notably, γH2AX wraps broken DNA ends along with hundreds of kb flanking DNA to facilitate the active turnover of the DNA damage sensor and repair proteins, and protects the damaged DNA ends from dissociation and degradation [54,55]. Given that the chromatin structure contains different topological borders, DSBs that disrupt these boundaries allow the γH2AX signal extension across either side of the breaking point; however, DSBs at or within the borders of a topological domain form a highly asymmetric γH2AX-regulated DDR platform [56]. Subsequently, MDC1 recruits the late performing effectors (within 5–60 min) such as ubiquitin ligases RNF8 and RNF168, which then promote the loading of BRCA1 and 53BP1 at DSB sites [57,58]. RNF168 mediates the ubiquitination of histone H2A at lysine 27, a major ubiquitin mark on damaged chromatin, in response to DNA damage. This chromatin ubiquitination then facilitates the recognition and recruitment of 53BP1, Rap80, RNF169, and RNF168 [59]. The reversal of chromatin ubiquitination is carried out by two E3 ligases, namely, TRIP12 and UBR5, promoting the ubiquitin-dependent degradation of RNF168 [60]. Like ubiquitylation, timely deubiquitylation is also important for the successful release of DSB repair factors, so that the next step of the repair process can occur. Deubiquitinase USP8 specifically removes the lysine 27-linked ubiquitin chain, which promotes the deacetylase activity of histone deacetylase 1 (HDAC1) [61]. Deletion of USP38 impairs the dissociation of NHEJ factors from DSB sites. Nucleosome destabilization around the DSB is actively conducted by histone acetyltransferase Tip60 and the ATPase activity of p400 in an MDC1-dependent but ATM-independent fashion. p400-mediated nucleosome destabilization is a key step toward RNF8-dependent ubiquitination of the 10 s kb of the chromatin region, and subsequent recruitment of 53BP1 and BRCA1 at the DSBs [62]. Furthermore, p400 ATPase also regulates the extent of AltEJ activity at the damaged chromatin. It has been reported that depletion of p400 significantly increased the frequency of AltEJ events by stimulating the recruitment of PARP1 and DNA ligase 3, leading to the deletion of large segments of chromosomes following DSB repair [63]. Like ubiquitylation, SUMOylation also plays an important role in chromatin remodeling during DDR signaling. It has recently been demonstrated that the interaction of TIP60 with DNA-PK catalytic subunit (DNA-PKcs) is crucial in making pathway choices in the S phase. The protein inhibitor of the activated STAT 4 (PIAS4) E3 ligase mediates SUMO2 modification of TIP60 at lysine 430 which attenuates its interaction with DNA-PKcs, thus promoting HR. The lysine 430 to arginine mutation of TIP60 suppresses HR, without affecting the NHEJ pathway, and abnormally increases DNA-PKcs phosphorylation at serine 2056 [64]. SUMO E3 ligase PIAS1 has been found to modulate the activity of polynucleotide kinase-phosphatase (PNKP), a damaged DNA-end processing enzyme, in response to the transcription-coupled DSB repair in genes associated with Huntington’s disease (HD) pathology. RIF1 is another DDR factor that plays a role in pathway choice between HR and NHEJ. During HR, BLM and RIF1 follow similar recruitment kinetics to stalled replication forks and form complexes [65]. However, RAD51 localization to damaged replication fork depends on its interaction with SUMOylated BLM [66]. Moreover, RIF1 SUMOylation by PIAS4 is critical for its interaction with 53BP1 to promote the NHEJ-mediated DSB repair at the G1 phase of the cell cycle [67,68]. In summary, different post-translational modifications (PTMs) of the DDR and repair proteins are essential for proper interaction modulating the chromatin landscape, to assemble an appropriate DDR platform for efficient DSB repair.

## 2. The Emerging Role of RNA/DNA-Binding Proteins in the DDR and DSB Repair

In the last decade, new research has identified several novel RNA/DNA binding proteins (RDBPs) as novel regulators of DDR and repair pathways. While many of these RDBPs have been demonstrated to possess well-defined roles in gene regulation and RNA metabolism, emerging studies have highlighted several non-canonical roles that include chromatin remodeling, DNA damage signaling amplification, and scaffolding of dynamic repair complexes at damaged chromatin. The heterogeneous nuclear ribonucleoprotein (hnRNP) family comprises a growing collection of RDBPs that have been extensively linked to nucleic acid metabolism. The hnRNP family was founded on the discovery of hnRNPs A/B and C, with later studies identifying a further 18 major protein members. Subsequent additions to the hnRNP family have generally been denoted by a letter to indicate the order of discovery or structural similarity to previously identified members; this naming convention is not universal, however, as other members such as TDP-43 and FUS lack both the hnRNP prefix and letter identifier [69,70]. While hnRNPs are classically recognized for their RNA binding activities, they have also been proven vital to successful DNA repair. In this section, we will critically review the mechanistic features of these non-canonical functions of RDBPs in the context of the DDR and repair. A summary of how and where important RDBPs interact in the DSB repair process is shown in Figure 1 with greater detail provided in Table 1.

### 2.1. The Diverse Roles of hnRNP Family Proteins in Multiple DDR and DSB Repair Pathways

DSB induction triggers the recruitment of dozens of damaged sensors and repair proteins to the break sites. A sudden increase in the local concentration of effector proteins at these sites accelerates the rate of DDR signal amplification and transduction to various parts of the cell, resulting in the rapid accumulation of repair pathway-specific proteins. New studies have demonstrated that these concentrated proteins coalesce into “membrane-less organelles” (MLO) due to their collective liquid–liquid phase separation (LLPS) properties. RDBPs have been reported to play critical roles in the formation and stability of these LLPS properties on chromatin [71,72]. Generally, proteins with low complexity domains (LCDs), such as the hnRNP family protein hnRNP-A1, have a greater propensity to form LLPS [73].

While there is extensive overlap in the functions of many hnRNPs between DDR signaling and DNA repair, some hnRNPs primarily affect the DDR signaling cascade. Specific damage recognition is critical for avoiding unnecessary activation by physiological DNA structures that mimic DNA lesions. Telomeres are characterized by TG-rich repeats of double-stranded DNA (dsDNA) that produce a unique secondary structure and terminate with a single-strand 3′ overhang [74,75]. These unique structures can be inadvertently recognized as DSBs by DNA repair proteins, resulting in the disruption of cellular homeostasis by inducing the DDR. Both hnRNP-A1 and -A2/B1 participate in telomere metabolism and prevent the telomeric DNA break-induced activation of the DDR. HnRNP-A1 accomplishes these effects by facilitating the association of telomerase enzymes with the 3′ telomeric ends, enhancing telomerase activation and promoting the formation of the Shelterin complex [76,77,78,79]. Phosphorylation of hnRNP-A1 by vaccinia-related kinase 1 (VRK1) and DNA-PKcs enhances its interaction with telomerase enzymes and accelerates the formation of the Shelterin complex, respectively [79,80]. Genome-wide mapping of the frequent DSB sites reveals a non-random distribution of DSB hotspot locations that are typically delimited into 50–250 kb DNA segments. Interestingly, nearly 30% of these hotspots contain clusters of coordinately expressing gene sequences with binding sites for PARP1 and hnRNP-A2/B1 [81]. HnRNP-A1 also regulates the transcription and alternative splicing of DDR-associated genes. Studies using oxaliplatin-treated HEK293 cells demonstrated that hnRNP-A1 and hnRNP-A2/B1 coordinate with other protein factors to drive DNA damage-induced alternative splicing of genes associated with apoptosis, cell cycle progression, and DNA repair [82]. Additionally, hnRNP-A2/B1 plays a direct role in DSB repair. Experiments using nuclear extracts of A549 cells demonstrated that hnRNP-B1 directly interacted and co-localized with DNA-PKcs following irradiation (IR) exposure. Notably, when hnRNP-B1 is ectopically overexpressed, a dose-dependent inhibition of DNA-PKcs’ phosphorylation activity is observed. As expected, the siRNA knockdown (KD) of hnRNP-B1 in ionizing radiation (IR)-exposed human lung cancer bronchial epithelial cells caused faster resolution of neutral comet assay tail moments when compared to controls [83]. These findings are clinically relevant because the overexpression of hnRNP-A2/B1 is observed in early-stage lung cancer and premalignant bronchial dysplasia, concomitant with the accumulation of unrepaired DSBs [84]. Despite the close structural similarities between hnRNP-A2/B1 and TDP-43 [85], their effects on DSB repair appear to be in direct opposition. 

Once the DDR is activated, it induces myriad effects on cell physiology designed to confer a survival advantage. One major part of this response is a global decrease in mRNA levels that coincide with DNA repair [86]. In the basal state, coding mRNA transcripts undergo cleavage at pre-mRNA 3′ ends and polyadenylation (pA) to confer stability and permit translation [87,88]. However, following the genotoxic stress disruption of pre-mRNA 3′ end processing can subvert this event and contribute to cell stress-induced transcriptional repression [89,90,91,92]. Yet, for DDR signaling and DNA repair to be successful, their respective components must be expressed, suggesting that a compensatory mechanism exists to allow their escape from damage-induced global suppression. To this end, hnRNP-A2/B1 exerts a multivariant recognition capacity conferred by its (RNA recognition motif) RRM1 motif for “AGG” and RRM2 motif for “UAG” sequences, thus playing an important role in the RNA matchmaker mechanism [93]. The hnRNP-F/H protein also contributes to this process by facilitating the expression of certain DDR genes, including p53. This is accomplished when hnRNP-F/H binds to the G4 forming a G-rich sequence in the 3′-UTR of the p53 pre-mRNA transcript. Under physiological conditions, this binding is accompanied by the concomitant recruitment of the cleavage stimulation factor (CstF) and poly(A) polymerase (PAP) leading to p53 translation. Following DNA damage induction, this binding activity is increased several-fold, thereby allowing p53 expression to escape downregulation [86]. Without this binding in hnRNP-F/H deficient cells, p53 expression is significantly impaired following UV exposure, leading to PARP1 cleavage and increased expression of the cellular senescence marker, p21 [94]. HnRNP-K also facilitates p53 expression during the DDR by enhancing its transcription and counteracting the human double minute 2 protein (HDM2)-mediated ubiquitination of p53. Among the targets of early DDR kinases, the phosphorylation of HDM2 and p53 disrupts their interaction and prevents p53 degradation [95,96]. At the same time, hnRNP-K undergoes damage-induced PTMs that prevent its HDM2-mediated degradation, and induce its translocation into the nucleus, where it promotes the transcriptional activation of p53 [95]. Multiple studies have reported that these effects were achieved by DNA damage-induced SUMOylation of hnRNP-K, secondary to the activation of several enzymes, including PIAS3 and polycomb protein 2 (Pc2) [97]. Importantly, these modifications are enhanced by the actions of the ATR kinase. Experiments using UV exposed cells treated with caffeine, an ATR kinase inhibitor, or siRNA KD of ATR, demonstrate a loss of both UV-induced hnRNP-K SUMOylation and its interaction with PIAS3. Analysis of hnRNP-K deletion mutants using pull-down assays revealed that the SUMOylation of the C-terminus confers increased stability by interfering with HDM2 binding, thus facilitating the mobilization of SUMO-hnRNP-K to the nucleus while leaving the N-terminal region free to interact with p53 [98]. A functional assessment reveals a loss of hnRNP-K delays or prevents cell cycle arrest, coinciding with decreased p53 and p21 expression [98]. hnRNP-L has also been linked to successful DNA damage repair. Following genotoxic stress, hnRNP-L is localized to damage sites, where it supports DSB repair by effectively recruiting early DDR factors such as ATM, 53BP1, and BRCA1 [99]. HnRNP-L depletion before oxaliplatin treatment causes significant decreases in the formation of 53BP1 and BRCA1 foci, while simultaneously increased staining for ATM phosphorylation at serine 1981 (pATM) and γH2AX foci, indicating persistent accumulation of unrepaired DSBs [99]. In vitro assays using NHEJ- and HR-specific reporter cell lines further demonstrate that the loss of hnRNP-L significantly impairs the efficiency of both pathways, and co-immunoprecipitation (co-IP) studies suggest that this may be secondary to decreased interaction with ATM, 53BP1, and BRCA1 [99]. 

In addition to being induced by DSBs, the p53 protein also participates in AltEJ repair, where it directly binds to ssDNA and dsDNA ends and exerts 3′–>5′ exonuclease activity [100,101]. Using extracts of p53 depleted cells incubated with DSB plasmids demonstrates an increased rate of nuclease-mediated degradation of both blunt and overhang DSB ends, suggesting that p53 facilitates AltEJ by preventing excessive DNA end degradation [102]. While many mechanistic details of this function remain elusive, it has been reported that hnRNP-G may facilitate p53 activity during DNA repair by protecting broken DNA ends from nuclease degradation [102]. Furthermore, p53 KD cells exhibit a decreased expression of hnRNP-G, with a three-fold reduction in AltEJ repair fidelity compared to controls. These effects are likely mediated by the direct interaction between damaged ssDNA or dsDNA ends and the RRM domain of hnRNP-G [102]. Following DNA damage, hnRNP-C appears to play a role in DSB repair pathway choice, where specific HR deficiencies are compensated by enhanced AltEJ activity. Tandem affinity purification and mass spectrometry analyses demonstrate that hnRNP-C interacts with the HR-associated PALB2/BRCA complex, and that hnRNP-C deficient cells exhibit significant decreases in expression of the HR-related proteins BRCA1, BRCA2, RAD51, and BRIP. Indeed, cross-linking immunoprecipitation (CLIP)-sequencing experiments confirm hnRNP-C’s binding to these transcripts at the expected sites and demonstrates the exonization of Alu elements within the same transcripts. Furthermore, these events are abrogated when cells are pretreated with RNaseA, confirming that the effect of hnRNP-C on HR repair is RNA-dependent [103]

### 2.2. Involvement of hnRNP-U in DSB Repair

Another critical aspect of the DDR involves the temporal order of repair pathways following genotoxic stress. HnRNP-U or Scaffold attachment factor A (SAF-A) is a 90 kDa protein, and the largest member of the hnRNP family. There are many roles ascribed to hnRNP-U, but among them is an increasing body of evidence linking it to the DDR and multiple DNA damage repair pathways [104,105,106]. Co-immunoprecipitation assays using FLAG-tagged hnRNP-U in HEK293 and U2OS cell lines treated with IR show a time- and phosphorylation-state-dependent interaction between hnRNP-U, Ku70, and DNA-PKcs [107]. The hnRNP-U protein is associated with modulating the balance between the NHEJ and BER pathways at the beginning of DNA repair. Following DSB induction, NHEJ-mediated DSB repair occurs within ~15–60 min, while BER occurs following successful NHEJ repair [107,108,109]. During this period, hnRNP-U undergoes DNA-PKcs-dependent phosphorylation, allowing preferential binding to Ku70, which favors the progression of NHEJ repair activities. Conversely, the interaction between BER factor NEIL1 and non-phosphorylated hnRNP-U enables NEIL1 binding to damaged chromatin and initiates BER after completing NHEJ. 

The hnRNP-U protein also plays a role in the two-phase dynamics of the association–dissociation process of DNA repair factors at the chromatin coupled with R-loop resolution [110]. The hnRNP-U protein binds to damaged chromatin in a PARylation-dependent manner; however, its dissociation from chromatin is mediated by ATM, ATR, and DNA-PKcs, thus restarting the ongoing transcription. These findings suggest an active DDR-linked anti-R-loop mechanism that excludes mRNA processing factors, such as hnRNP-U, TAR DNA binding protein 43 kDa (TDP-43), and Fused in Sarcoma (FUS), from the damaged transcribed sites in the chromatin. Similarly, hnRNP-D has been found to regulate R-loop resolution following DNA damage. Studies suggest that hnRNP-D accomplishes this function by directly coordinating with hnRNP-U at damaged sites. Notably, loss of hnRNP-D not only results in increased R-loop accumulation but also the failure of hnRNP-U to colocalize at damage sites [111]. Furthermore, hnRNP-D depletion has been linked to impaired HR-mediated DNA DSB repair by blockade of DSB end resection.

### 2.3. Involvement of TDP-43, FUS, and RBM14 in DSB Repair

Neurodegeneration-associated RDBPs TDP-43, FUS, and RBM14 have recently been implicated in DDR and DSB repair mechanisms for their direct roles in modulating genome integrity and fidelity. Previous studies have indicated that these proteins regulate DNA repair mechanisms in association with HDAC1, a critical chromatin modifier, the depletion of which can lead to genotoxic stress in neurons [112,113,114,115]. The hnRNP proteins, TDP-43 and RBM14, have both been associated with direct roles in NHEJ-mediated DSB repair. 

TDP-43 is a 414 amino acid protein of the hnRNP family encoded by the TARDBP gene. It was first discovered as a transcriptional repressor targeting the TAR DNA sequence of human immunodeficiency virus 1 (HIV-1) [116] and is highly conserved across multiple species, including humans, mice, drosophila, and C. elegans. It is ubiquitously expressed, with its RNA detected in all tissues, but most strongly expressed in CNS, endocrine, muscle, and gastrointestinal tissues (proteinatlas.org v20.1). Structurally, TDP-43 consists of two RNA RRMs, a prion-like glycine-rich domain, bifurcated nuclear localization sequences (NLS), and nuclear export sequences (NES) [85]. Among the hnRNP family of proteins, it closely resembles hnRNP A1 and A2/B1 [117], but is unique for its combined RNA and DNA binding capabilities. TDP-43 has been implicated in myriad roles related to RNA metabolism, including transcriptional repression, pre-mRNA maturation, alternative splicing, micro-RNA biogenesis, interaction with long non-coding RNA, and even autoregulation of its transcription [85]. 

The possible role of TDP-43 in DNA damage repair was first indicated by a proteomic study, which identified a significant interaction between TDP-43 and Ku70, one of the apoenzymes critically involved in classical NHEJ repair [118]. A subsequent investigation by Hegde and colleagues expounded on this finding, and for the first time, we demonstrated the direct role of TDP-43 in NHEJ-mediated DSB repair [119]. At baseline, TDP-43 depleted neuronal cells exhibit increased genomic instability, increased expression of pro-apoptotic factors, and persistent DSB accumulation over time, without any external DSB inducing agent. Using neutral comet assays and kinetics of DSB foci disappearance via live-cell imaging, DSB-inducing etoposide or bleomycin treatment of TDP-43 depleted cells demonstrate ~10-fold higher tail moment and slower disappearance of 53BP1 foci compared to controls. Furthermore, induction of DSBs leads to an enhanced association between TDP-43 and DDR factors γH2AX, pATM, and p53BP1, and NHEJ proteins Ku, DNA-PKcs, DNA polymerase lambda (Polλ), XRCC4, and DNA Ligase 4. This study also revealed that TDP-43 neither associates with the XRCC1/DNA Ligase 3 complex nor dynamically interacts with DNA polymerase µ (Polµ) in response to DSB induction; this underscores the specificity of TDP-43′s association with XRCC4/DNA ligase 4 and Polλ [119]. 

At the micro-irradiated DSB track, TDP-43 follows the recruitment kinetics of Ku70 and remains at the site until the completion of the repair process, a fact further supported by TDP-43′s critical scaffolding role in the recruitment of DNA end ligation complex (XRCC4/DNA Ligase 4), the rate-limiting step of NHEJ, to the DSB sites. Subsequent in vitro biotin-affinity co-elution experiments have demonstrated TDP-43′s affinity toward the free DSB end [119]. It is important to mention in this context that the loss of TDP-43 neither inhibits DDR activation nor prevents the formation of the DSB ligation complex consisting of XRCC4, XRCC4-like factor (XLF), and DNA Ligase 4; however, the impaired DSB ligation step, in turn, leads to the accumulation of unrepaired DSBs and simultaneous hyperactivation of DDR signaling, resulting in chronic inflammation and cell death. Furthermore, a mutant TDP-43 that mislocalized to the cytosol traps XRCC4 and DNA Ligase 4 complex, which may prevent their translocation from the cytosol to the nucleus following genomic DSB induction [120]. 

Another RDBP protein, FUS, has also been reported to modulate the recognition and repair of DSBs in neurons [114]. Although this study highlights the role of FUS in the formation of post-damage γH2AX and 53BP1 foci in cells, the interactions of FUS with classical NHEJ repair factors have also been demonstrated in this study. Wang et al. reported that FUS was essential for the PARylation-mediated activation of the XRCC1/DNA Ligase 3 complex in response to oxidative DNA damage [121]. A recent proteomics interactome study revealed that the levels of TDP-43 and FUS in the cell were not dependent on each other, and they often shared common interacting partners [122]. Some of the first characterizations of FUS in knockout (KO) mice models discovered that the genetic deletion of either the zinc finger motif of the 8th exon human homolog in mice resulted in perinatal death and impaired fertility; cell lines derived from these models exhibited increased genomic instability and sensitization to IR [123,124]. Its effect on fertility prompted the discovery of its contribution to homologous recombination, wherein it promotes the formation of DNA D-loops and the annealing of homologous DNA [125]. Finally, FUS participates in stress granule (SG) formation following genotoxic stress. This key step in the cellular defense against genotoxic stress occurs in parallel with DDR activation, and largely functions to protect long mRNA/pre-mRNA transcripts. Abrogating SG formation cell cultures results in enhanced apoptotic cell death after genotoxic stress, suggesting an essential role in cell survival [126]. Interestingly, both TDP-43 and FUS have been recognized as important factors for efficient SG assembly/disassembly dynamics in response to cellular stresses [127,128].

RBM14, also known as the RRM containing coactivator activator (CoAA), is highly expressed in early embryonic stem cells and has been implicated in DNA repair and cell proliferation in cancer cells [129]. RBM14 is primarily known for its regulatory role in RNA metabolism; however, some reports have linked it to the cell stress response, where it joins other RDBPs such as FUS, as a component of stress-induced nuclear paraspeckles [130,131,132,133]. Although the function of paraspeckles continues to be the subject of investigation, some components have been associated with the DDR, especially FUS and NONO [134,135]. RBM14 is also involved in DDR and DNA repair. Glioblastoma cell lines lacking RBM14 demonstrate enhanced sensitivity to IR, and evidence of unrepaired DSB accumulation concomitant with decreased levels of phosphorylated DNA-PKcs and NHEJ repair efficiency [129]. Recent evidence demonstrates that RBM14 plays a critical role in the execution of the NHEJ process. Using HEK293 cells expressing DSB reporter constructs, the loss of RBM14 was demonstrated to drastically increase the activity of mutagenic NHEJ repair [136]. The DNA deep sequence analysis of the DSB repair junctions demonstrates that RBM14 KD cells contain a seven-fold decrease in reads exhibiting faithful DSB repair and a significant increase in reads containing microhomology signatures [136]. Interestingly, the dissociation of the Ku protein complex from damaged ends appears to require RBM14 and is necessary for the progression of the NHEJ pathway [136]. The loss of RBM14 nearly triples the time needed for the Ku protein complex to dissociate, and DNA Ligase 4 to bind to damaged sites [134]. Taken together, RBM14 appears to facilitate NHEJ-mediated DSB repair by promoting the dissociation of the Ku70/80 heterodimer from DNA ends and/or the docking of DNA Ligase 4 complex at the DSB sites.

## 3. DNA Damage and Its Pathological Consequences in Neurological Disorders

The formation of DNA damage and its downstream signaling are increasingly recognized as fundamental contributors to neurodegeneration and brain aging. While it is known that unrepaired DNA DSBs is uniquely toxic to vulnerable long-lived cells of the CNS, the exact mechanisms linking DNA repair defects to neuronal loss are incompletely understood. One potential consequence of unrepaired DNA breaks may involve post-mitotic neurons re-entering the cell cycle, resulting in apoptosis [137]. Alternatively, unrepaired DNA damage may induce neurons to adopt a senescence-like phenotype characterized by dysfunctional neurophysiology, metabolic dysregulation, and secretion of harmful senescence-associated molecules. Moreover, neurons harboring unrepaired DNA breaks with accompanying nuclear membrane damage may cause nuclear DNA-mediated activation of the cytosolic cyclic GMP-AMP Synthase (cGAS)- Stimulator of Interferon Genes (STING) pathway [138,139]. The STING pathway involves the TANK Binding Kinase 1 (TBK1)- Interferon Regulatory Factor 3 (IRF3) signaling-mediated activation of the innate immunity to sustain the chromosomal stability in a p21-depedent manner. Activation of this pro-inflammatory pathway may contribute to the development of a proinflammatory milieu surrounding neural tissues altering the cellular function. In either case, the neuronal activity becomes abnormal, and if left unresolved, persistent inflammation may lead to apoptosis and irreversible tissue loss. For these reasons, DNA damage is critically positioned to help explain how neurons change with aging, respond to environmental exposures, and contribute to genetic neuropathological phenotypes.

Furthermore, initially discovered in both the normal and cancer cell lines, extrachromosomal circular DNA (eccDNA) has been identified to cross-talk with STING-associated pro-inflammatory pathways [140,141,142]. The eccDNA structure is derived from transcriptionally active, exon rich, and non-repetitive DNA sequences [143]. Interestingly, eccDNA production involves neither NHEJ nor HR machinery. Instead, mismatch repair factor MutS Homolog 3 (MSH3) has been implicated in regulating the cellular load of eccDNA molecules [143]. Some have suggested the origin of eccDNAs in post-mitotic neurons may be associated with R-loop dysregulation at transcriptionally active genomic regions [144]. However, in another study by Zhu et al. [145], eccDNAa were produced from DNA DSBs flanking short microhomology sequences, suggesting critical roles of NHEJ and MMEJ in post-mitotic neurons. To date, most studies examining the functions of eccDNAs have focused on their multi-faceted, regulatory roles in the cell, including apoptosis, aging, and neurodegenerative diseases [141]. Additional investigation will be required to elucidate the relationship, if any, between eccDNA production and function with members of the hnRNP family of proteins and their disease-relevant mutations [146]. In the following sections, we describe the contributions of DNA damage and DSBs to common motor neuron diseases, dementias, and finally brain aging.

### 3.1. DNA Damage in Motor Neuron Disease

Motor neurons are specialized cells that relay electrochemical signals from the brain to the musculoskeletal system to facilitate smooth, controlled voluntary limb movements. Motor neurons and their respective diseases may be subdivided into two groups: upper motor neurons (UMNs) and lower motor neurons (LMNs). UMNs are glutamatergic neurons with cortical somas and axonal projections that extend inferiorly through the brainstem and spinal cord to synapse onto LMNs. These LMNs in turn have somas in the ventral spinal cord with distal cholinergic projections terminating at the neuromuscular junction (NMJ) of skeletal muscles. Worldwide, the burden of motor neuron diseases (MNDs) is growing at an alarming rate. Approximately 331,000 people suffer from MNDs, ultimately resulting in 34,325 deaths in 2016 alone. To date, dozens of pathology-associated genes and related signaling pathways have been linked to the ages of symptom onsets and rates of disease progressions for several MNDs. In this section, we survey selected genetic factors associated with selected MNDs and highlight their contributions to genome instability and/or defective DDR signaling.

#### 3.1.1. Amyotrophic Lateral Sclerosis (ALS)

Perhaps the most widely recognized motor neuron disease is ALS or Lou Gehrig’s Disease. ALS is a devastating disease that usually begins around the fifth decade of life and has an incidence rate between 1.5 and 2.7 per 100,000 person-years in Europe and North America [147,148,149,150]. The hallmark clinical feature of ALS is the combination of upper and lower motor neuron signs; the classical presentation involves asymmetric spasticity and paresis, beginning with the distal limbs. The disease progression is relentless with median survival from time of symptom onset limited to just 3–5 years. The proximal cause of death in most patients is respiratory failure and cardiac arrest [151]. Although ALS predominates in the adult population, juvenile ALS cases have also been reported. Juvenile ALS patients progress similarly to that adults, but while the genetic origins of adult-onset ALS are commonly sporadic, juvenile cases are remarkable for their association with discrete, inherited mutations. Several of these genes participate in DNA repair, DDR signaling, reactive oxygen species (ROS) production, and neuroinflammation. Although more than a dozen genes have been implicated in ALS pathogenesis, some of the most studied mutations involve *TARDBP*, *FUS*, *C9ORF72*, syntaxin (*SETX*), and ataxin 2 (*ATXN2*), TANK-binding kinase 1 (*TBK1*), and matrin 3 (*MATR3*) genes [85]. 

SETX acts as an essential R-loop-associated helicase for the resolution of DNA:RNA hybrids that are formed when a template DNA strand hybridizes with its nascent RNA transcript [152]. SETX acts by removing R-loops flanking sites of DNA DSBs in actively transcribed genes by recruiting RAD51 to prevent aberrant chromosomal translocation. As expected, SETX depletion enhances γH2AX at DNA break sites. Although SETX has not been specifically marked to enhance DSB repair or mediate DSB end resection, it is known to modulate cross-talk between replication stress-induced DDR activation and the unfolded protein response via the PERK/ATF4 signaling axis [153]. SETX itself is also under the regulation of multiple factors, including ubiquitin-specific peptidases that may play a role in neurodegeneration [154]. Several pathogenic SETX mutations have been reported linking SETX proteinopathy with juvenile-onset ALS [155,156]. 

Mutations in TDP-43 and FUS have also been extensively linked to the pathogenesis of ALS. Mutations in TDP-43 have been identified in up to 95% of sporadic ALS cases, and mutations in both TDP-43 and FUS have been linked to rare familial forms of ALS [157]. The nucleo-cytoplasmic mislocalization of mutated TDP-43 and FUS is a hallmark feature of ALS. Both TDP-43 and FUS are concentrated in the nuclei of healthy neurons; however, in ALS-affected tissues, both proteins accumulate in the cytosol as intranuclear inclusion bodies. The current view regarding this mislocalization is that it leads to a simultaneous loss of function in the nucleus and a gain of toxicity in the cytosol [85]. As previously discussed, TDP-43 and FUS play critical roles in the resolution of DNA damage, and their pathological nuclear loss causes increased genomic instability. Loss of nuclear TDP-43 and FUS also results in substantial alterations to mRNA processing and gene expression. Previous studies have demonstrated that FUS alone affects the processing of at least 5500 RNA targets with TDP-43, likewise demonstrating a significant overlap [158,159]. One major function of TDP-43 and FUS in RNA metabolism is the mediation of pre-mRNA splicing. For example, the P525L mutation in FUS causes nuclear clearance with subsequent inhibition of select intron splicing events due to poorly-localized spliceosome components [160]. TDP-43 also exerts multiple effects on RNA processing with consequences for motor neuron regenerative capacity. Studies have demonstrated that TDP-43 regulates the expression of neuronal growth factor, stathmin-2. In vitro experiments have revealed that TDP-43 binds to the first intron of stathmin-2 pre-mRNA where it prevents access to a cryptic poly-adenylation site by other RNA processing factors. Without TDP-43, the retention of this site results in a truncated mRNA product that disrupts axonal regeneration in motor neuron cell cultures [161]. Interestingly, TDP-43 has also been linked to ALS pathology outside the context of inherited mutations. A recent investigation identified cell stress-induced alternative splicing of TDP-43 itself as producing a shortened splice variant with decreased cytoplasmic solubility and a propensity to sequester full-length TDP-43 in cytosolic granules [162,163]. These findings suggest that exogenous influences, such as neuronal activity, may contribute to TDP-43 pathology, and may help explain the role of TDP-43 in sporadic ALS disease. Finally, the effects of mutated TDP-43/FUS in ALS may synergize with mutations in other ALS-associated proteins. For example, ALS-linked mutations in MATR3, an RNA binding protein, were demonstrated to disrupt global nuclear mRNA export that specifically includes TDP-43 and FUS transcripts [164].

The *C9ORF72* gene mutation is another well-described genetic cause of ALS. This mutation consists of an intronic hexanucleotide expansion repeat within the *C9ORF72* gene. While the function of the C9orf72 protein is unknown, its mutation is the most common genetic cause of familial ALS [165]. Neurons carrying the mutation exhibit cytoplasmic mislocalization of TDP-43, cytoplasmic aggregates, transcriptomic abnormalities, and a marked elevation in markers of DNA DSBs [85,166]. One way in which the mutation achieves these effects is through the production of expansion-encoded dipeptide repeat proteins. One recent study demonstrated that the expression of these peptides could slow DSB repair by decreasing the efficiency of NHEJ, single-strand annealing, and MMEJ pathways [167]. The C9orf72 expansion mutation has also been demonstrated to promote persistent R-loop structures with resulting increases in R-loop-mediated DSBs in mutant cell lines [168,169]. 

ATXN2 is another RNA binding protein classically associated with autosomal dominant spinocerebellar ataxia type 2 (SCA2), but also confers an increased risk of developing ALS. Wildtype ATXN2 is a cytosolic protein that primarily facilitates RNA metabolism. While the expansion of repeat mutations exceeding 35 repeats is linked to SCA2, intermediate repeat mutations, between 24 to 34 repeats, confer three-fold greater odds of developing ALS [170]. Mutated ATXN2 is believed to increase the risk of ALS by promoting stress granule formation and cytosolic sequestration of TDP-43 [171].

#### 3.1.2. Other Selected Motor Neuron Diseases

Primary lateral sclerosis (PLS) and ALS are often regarded as pathological members of a spectrum of motor neuron diseases. Indeed, both PLS and ALS share many characteristics; however, PLS is rare in comparison and is defined by its restriction to UMNs [172,173]. Although relatively few studies have examined the pathology of PLS, one study of seven PLS patients revealed extensive TDP-43 positive inclusions in cortical and corticobulbar tract neurons. As expected, few inclusions were identified in LMNs [174]. Interestingly, this study and others have noted significant concordance between the extent of UMN TDP-43 inclusions with comorbid frontotemporal atrophy and abundant cortical TDP-43 pathology [175,176]. These results suggest the location of TDP-43 pathology likely plays a decisive role in determining disease phenotype. 

Spinal muscular atrophy (SMA) is the leading cause of pediatric neurodegenerative disease and affects 4 to 10 per 100,000 live births annually [177]. SMA results from the mutation or loss of the survival motor neuron 1 (*SMN1*) gene. Without SMN1, motor neurons exhibit degenerative changes leading to progressive diffuse proximal weakness and hypotonia that often leads to premature death [178,179]. While much is unknown regarding the pathways connecting the loss of SMN1 to motor neuron degeneration, DNA DSBs have been reported as a possible contributor. In healthy neural tissue, SMN1 co-localizes with SETX and DNA-PKcs in subnuclear bodies. Experiments using patient-derived fibroblasts and spinal cord tissue have reported that a loss of SMN1 causes decreased expression of SETX and DNA-PKcs with a concomitant increase in unresolved R-loop structures, DSB markers, and DDR activation [180]. These findings demonstrate that persistent DNA damage and DSB accumulation via unresolved R-loop structures likely contribute to motor neuron degeneration in the absence of SMN1. 

Machado–Joseph disease (MJD) is the most common cause of spinocerebellar ataxia worldwide. The hallmark clinical features of MJD include progressive ataxia and prominent cerebellar signs, although clinical manifestations can vary widely and often include pyramidal signs with peripheral amyotrophy [181]. These signs result from the degeneration of multiple CNS systems, including the cerebellum and spinal cord due to CAG expansion repeats within the *ATXN3* gene [182]. The resulting expansion repeat proteins lack their native function and contribute to characteristic intraneuronal inclusions that aggregate with other cytosolic proteins. Recent studies have demonstrated that native ATXN3 normally associates with polynucleotide kinase 3’-phosphatase (PNKP), a DNA repair protein required for processing broken DNA ends, RNAPII, in addition to major NHEJ repair proteins. Subsequent in vitro experiments demonstrate that KD of ATXN3 abrogates the activity of PNKP and decreases the rate of error-free DSB repair of linearized reporter plasmids. Furthermore, brain extracts from MJD patients and transgenic mice confirmed diminished PNKP activity with simultaneous increases in phospho-53BP1 expression and markers of DNA strands [183,184].

### 3.2. Dementia-Associated Neurodegenerative Diseases

Dementia is a disorder characterized by a decline in cognitive abilities across at least one cognitive domain, such as learning and memory, executive function, apraxia, and aphasia, according to ICD 10 criteria [185]. As the US population continues to age, the overall burden of dementia has likewise increased, and this trend is expected to continue over the coming decades. Unlike mild cognitive impairment (MCI), dementia is generally irreversible, more severe, and not considered a normal aspect of aging. The two most common causes of neurodegenerative dementia include AD and frontotemporal dementia (FTD), followed by dementia with Lewy bodies and PD dementia [186,187,188]. 

FTD describes a spectrum of neurological disorders affecting distinct brain networks that result in characteristic clinical and neuropathologic patterns. It is one of the more common causes of early-onset dementia (i.e., ages < 65 years), where it is observed with roughly the same incidence as AD [189]. FTD can be distinguished from AD by clinical signs of altered personality, behavior, and language preceding memory loss. Like other neurodegenerative diseases, neuropathological findings include macroscopic atrophy of frontal and temporal lobes with microscopic findings of gliosis, neuronal apoptosis, and prominent neural inclusion bodies. The two most common proteins found in these inclusion bodies are tau and TDP-43; other proteins, such as FUS, are also found to a lesser extent [190]. Mechanisms underlying the toxic effects of tau aggregates in neurons remain uncertain; however, the combined loss of function and gain of toxicity model is likely relevant. Conversely, the pathological contributions of TDP-43 aggregates in FTD are much the same as those previously discussed in motor neuron diseases. Like ALS, the genetic origins of FTD are mostly sporadic with monogenic cases comprising just 20% of cases. Known genetic causes of FTD include microtubule-associated protein tau (MAPT), C9orf72, TDP-43, FUS, TBK1, and others [165]. As previously discussed, mutations in TDP-43 and FUS are closely associated with genomic stability and have been implicated in neurodegeneration [191]. Repeat expansion mutations in C9orf72 have also been described in FTD [192]. Interestingly, recent work by Gitler and colleagues identified p53 as a central factor in driving C9orf72-mediated neurodegeneration. Neurons expressing the dipeptide repeat translated from C9orf72 expansion mutations promote neurodegenerative changes via broad transcriptional alterations that require a functional p53 transcription factor. Remarkably, the ablation of p53 in a mouse model of C9orf72 ALS/FTD completely reversed the neurodegenerative changes, and increased survival [193]. These results suggest that neurodegeneration in ALS/FTD likely proceeds from complex transcriptional reprogramming regulated by the integrated influences of the cell stress response, DNA strand breaks, chromatin remodeling, and the DDR.

Alzheimer’s disease is the most common cause of dementia in older adults, affecting an estimated 47 million people worldwide [190,194]. Age is the strongest risk factor for AD, with patients rarely becoming symptomatic before the fifth decade. AD is characterized by the insidious onset of impaired higher cognitive functions followed by progressive deficits in memory, judgement, personality, language, and mobility. Over 5–10 years, most patients become profoundly disabled [190]. Neuropathological hallmarks of AD include intraneuronal aggregates of amyloid-beta (Aβ) plaques and neurofibrillary tau tangles. Aβ plaques are pathognomonic for AD, and the current view regarding these inclusions suggests Aβ generation may serve as an initiating event for AD through various mechanisms including defective autophagy and mitophagy [195]. Consistent with this view, emerging evidence suggests DNA DSB repair may play an important part in the pathogenesis of AD. Analysis of post-mortem human brain tissue has demonstrated excessive DNA damage associated with AD at all stages of the disease, along with simultaneous alterations in the expression of repair factors [196,197,198]. Some reports indicate that the burden of DSBs may specifically predict clinical severity, as the staining of human cortical tissues for DSB marker γH2AX and DDR factor 53BP1 inversely correlate with clinical measures of cognitive function [198]. The findings of positive γH2AX staining in human postmortem and transgenic mouse tissues have been corroborated by multiple studies. Furthermore, DSB markers are usually observed in neurons and astrocytes, but not in oligodendrocytes, of the hippocampus and frontal cortex of AD and MCI tissues [199,200], which is consistent with the expected AD pathology. The source of AD-related DSBs remains uncertain. One explanation involves the dysregulated expression and activity of DNA repair enzymes. Given the prominent role of NHEJ in post-mitotic neuron DSB repair, it follows that disruptions in this pathway may promote DSB accumulation. However, evidence supporting this notion is limited. One retrospective study of postmortem human tissues demonstrates a decreased expression of DSB sensing factors Ku70/80 heterodimer and an insignificant trend in the decreased expression of DNA-PKcs [201]. In subsequent in vitro experiments, it was observed that PC12 cells treated with exogenous Aβ demonstrate decreased DNA-PKcs expression with expected increases in DSB accumulation. This result is not recapitulated in APP/PSEN1 transgenic mouse models, however. Alternatively, these tissues demonstrate decreased expression of RAD51, an HR-associated repair factor [202,203]. HR has also been implicated in AD pathogenesis by other studies. Notably, the tumor suppressor breast cancer susceptibility gene 1 (BRCA1)-ATM kinase signaling axis plays a crucial role in the development of brain function and size by modulating the polarization of neural progenitor cells (NPC) [204]. Given the essential role of BRCA1 in the HR-mediated DSB repair [205] and exploitation of this pathway by NPCs toward neuronal plasticity related to cognitive functions [206], the BRCA1 pathology seems more relevant to the hippocampal and entorhinal cortex neurons as well as migratory NPCs from the subventricular zone in the AD and dementia. Further research is needed to understand whether this mechanism can also affect the motor neurons, and if yes, then to what extent. Furthermore, the measurement of neuron-specific DNA methylation patterns across multiple brain regions identified hypomethylation of the *BRCA1* gene promoter region, with subsequent increased protein expression in AD neurons of the hippocampus, cerebellum, and occipital lobes. Although these regions also exhibit tau pathology, only the hippocampus exhibits significant DSB burden and cytosolic mislocalization of insoluble BRCA1 aggregates. Hippocampal BRCA1 aggregates strongly co-localize with both tau and Aβ inclusions, suggesting that Aβ may contribute to AD-related DSBs. In vitro experiments using human amyloid precursor protein (APP) overexpressing N2a swe.10 cells, N2a cells treated with exogenous Aβ, and N2a cells co-incubated with N1a swe.10 cells, all demonstrate increased cytosolic BRCA1 proteins. Lentiviral transfection of BRCA1 specific shRNA in N2a swe.10 cells significantly increases the DSB burden [207]. Although these findings point toward a causative role for BRCA1 and HR repair in AD-linked DSBs, other studies have reported conflicting results. Studies using Chinese Hamster Ovary (CHO) cells overexpressing APP and Aβ demonstrate persistent increases in DSB accumulation with decreased expression of BRCA1 protein, but not MRN or 53BP1. In the same study, hippocampal neurons of transgenic mice demonstrate no change in expressions of BRCA1, MRN, or 53BP1 protein or mRNA despite the presence of DSBs. Additionally, the qRT-PCR analysis of postmortem human tissues demonstrates decreased BRCA1 and MRN transcripts, but not for 53BP1, compared to controls [200]. Collectively, these data support the role of HR repair proteins in the pathogenesis of AD-related DSBs, although additional work is needed to elucidate underlying mechanisms. Moreover, in light of recent evidence linking Aβ plaques with other neurologic diseases such as autism [208], it is all the more interesting that these data suggest a causative role of Aβ in neuronal DSB induction. Aberrations in the DDR have also been linked to AD-related DSBs. Similar to that observed with BRCA1, postmortem human cortical neurons exhibit combined evidence of DSBs with cytosolic aggregates of p53 that strongly co-localize with tau inclusions. In the presence of DNA damage, activated phospho(p)-p53 levels are expected to increase and translocate to the nucleus to activate specific DDR pathways. In both human AD and transgenic mouse tissues, however, p-p53 remains trapped in the cytosol. Importantly, DSB-related downstream targets of p53 are also disrupted, including decreased levels of acetylated K382 p53, and p53-inducible ribonucleotide-reductase small subunit 2 [209]. Hence, these findings indicate neuronal DSBs are closely linked to AD pathogenesis from its earliest stages. Whether DSBs are a cause, or an effect of AD pathology remains uncertain. It is clear, however, that the neuron’s ability to manage DSBs in the context of AD is severely compromised, and that targeting DNA repair or DDR signaling proteins may prove useful as therapeutic strategies.

### 3.3. Brain Hemorrhage and Associated Neurological Consequences

Intracerebral hemorrhage (ICH) is the second most common cause of stroke and carries a significant risk of morbidity and mortality. ICH may occur secondary to several mechanisms, but is defined by the leakage of blood from the vasculature either into or around the brain parenchyma; bleeding usually happens in the anterior and posterior regions of subcortical territory [210]. In an acute setting, ICH poses an immediate danger as the accumulation of blood within the cranial space exerts pressure on the brain and reduces cerebral perfusion. In chronic phases, neural toxicity mediated by the effects of extravasated blood and its breakdown products leads to excitotoxicity and oxidative stress [211]. These effects are thought to develop secondarily to the exposure of neural tissue to blood-derived products (BDP), such as iron, which is a prooxidant [212]. Erythrocyte hemoglobin, an oxygen-transporting protein, carries four heme prosthetic groups amounting to four atoms of iron and is the source of iron toxicity following ICH. The breakdown of erythrocyte corpuscles begins within minutes from initial contact with brain parenchyma. Over the ensuing days to weeks, the slow release of BDPs from the hematoma saturates local tissue where its cytotoxic effects are observed [213]. Hemoglobin, when outside of the erythrocyte, is oxidized to methemoglobin, which then dissociates into dimers. With time these intermediates form hemichromes that break down into heme, which is finally catabolized by heme oxygenase to release free iron ions [214]. During this process, heme released from Hb retains its oxidizing characteristics and directly leads to the overproduction of ROS, oxidative stress, inflammation, and tissue damage [215]. Free radicals in the released ROS act to indiscriminately modify and destroy membranes, lipids, proteins, and nucleic acids that disrupt cellular and organ function [216]. Oxidative stress-mediated genomic damage and instability observed in neural tissue following ICH are also attributed to both hemin and free iron [217]. Importantly, some studies demonstrate that hemin has DNA strand cleavage activity in plasmid DNA [218], which may result in widespread induction of SSBs [219]. Hegde and colleagues recently demonstrated that hemin induced DNA DSBs in both the nucleus and mitochondria of treated cells. Notably, the DDR-mediated senescence-like phenotype adopted by these cells also appeared to confer resistance to iron-mediated ferroptosis, which was likely a critical adaptation for safe degradation/detoxification of hemin [220].

Heme products released after ICH also have been demonstrated to upregulate pro-inflammatory markers, including IL-6. Increases in IL-6 are credited with activating NF-kB inflammatory signaling pathways and driving phosphorylation of STAT3, which is associated with the induction of mediators of iron metabolism, such as hepcidin [221,222]. Notably, hepcidin is also associated with chronic cognitive impairment in ICH survivors [223]. Hepcidin has been detected in both serum and brain tissue following ICH. In serum, hepcidin binds the iron exporting channel protein, ferroportin, which inhibits iron efflux from microvascular endothelial cells and macrophages. However, some studies have reported elevated hepcidin in neurons of the brain exacerbates oxidative injury [224]. 

Free iron, the second major constituent of BDPs, is typically distributed in the basal ganglia, thalami, and white matter following intraparenchymal bleeds [225]. Nonheme iron is capable of generating ROS that causes oxidative brain damage via a process termed ferroptosis [226]. Iron is reported to affect the neural genome in two ways: the first is through direct oxidative damage of the DNA molecule itself, and the second is by oxidation of repair proteins such as NEIL1. In this manner, the iron component of BDPs acts to directly damage DNA integrity while simultaneously inhibiting DNA repair [227]. In many cases, iron exposure is insufficient to directly cause cell death but instead induces a senescence-like phenotype via activation of DDR signaling [220,228]. These senescence-like cells may express ROS-associated markers of cell cycle arrest, such as p21 [229], and may exhibit altered morphology characterized by an enlarged, aberrant organellar structure. Importantly, some reports suggest that this senescence-like state may confer resistance to ferroptosis-mediated cell death [230]. While telomere shortening is a considered the classic mechanism for cellular senescence, others have shown post-mitotic cells/tissues may undergo telomere length-independent damage, which may then lead to a senescent state via a non-canonical, senescence-linked, pro-hypertrophic, and pro-fibrotic secretory phenotype [231,232]. It remains an open-ended question how the senescence mechanism in post-mitotic cells can be accelerated by the biological aging processes [233].

Hegde and colleagues have reported a connection between such a senescence-like state and ferroptosis during ICH pathology. Even minimal concentrations of hemin can induce the formation of DSBs that cause senescence-like changes in 40% of cultured endothelial and neuronal cells. Furthermore, these cells display resistance to cell death after adding cytotoxic concentrations of iron compared to the controls [228]. These findings suggest that hemin exposure induces stress-related adaptations that permit survival after exposure to otherwise lethal doses of iron.

### 3.4. DSB Damage in the Aging Brain

In 2018, the World Health Organization declared aging a cause of disease [234]. In response to this new status, researchers and clinicians shifted their study of this universal condition to better understand the underlying effects of aging. The effects of aging are widespread throughout the body; however, of particular note to this review is the study of the aging brain. Each day the neuronal genome is under constant assault by deleterious factors derived from environmental and physiological origins [235]. The resultant molecular damage inevitably leads to genomic instability and progressive degradation of the genetic blueprint. These changes in the structure and composition of genetic molecules may help explain why organisms become increasingly vulnerable to disease with increased mortality as they age. This so-called DNA damage theory of aging is one of many developed to explain how organisms predictably and naturally change over time. 

The effect of DNA damage is especially relevant to the brain because its nondividing neuronal cell population must survive for the duration of an organism’s life with limited capacity for self-renewal. Consistent with the DNA theory of aging, some studies have observed an increased frequency of positive γH2AX cell staining in multiple tissues from aged animals, including the brain. The γH2AX is widely recognized as an indicator of DSB presence in the genome. Its increased persistence in aged animals, therefore, suggests an accumulation of unrepaired DSBs and implicates γH2AX as a molecular marker of aging [236,237]. While the exact reasons for this increase are likely multifactorial, studies from the early 2000s clearly demonstrate that aging neurons exhibit a diminished capacity for DSB repair and that the remaining capacity for repair is driven primarily by the error-prone NHEJ [238,239].

One mechanism by which DSBs may contribute to brain aging is through the conversion of damaged neurons into a senescence-like state. Because DNA DSB is a particularly lethal form of damage, the cellular response to their development is swift and complex. The resulting DDR induced by DSBs can either lead to repair of the lesion, initiation of apoptosis, or induction of a senescence-like phenotype [229,232,233,240,241]. The mechanisms by which DSBs are repaired in neurons may also vary in their fidelity to the original sequence, thus introducing new mutations with uncertain consequences. The choice of the repair pathway also influences the outcome of DSB repair and is likely determined by numerous factors, such as type and number of DSBs, among others [242]. When chronic, these senescence-like changes can occur in nearly every CNS cell type and have been attributed to functional aberrations found in both healthy aging and neurodegenerative diseases [243,244,245,246]. It is believed that the continuous activation of the DDR produces these changes in neurons [247]. The choice of cell fate following the induction of DDR depends largely on the duration and severity of the DNA damage. Post-translational modifications of various effector proteins within the DDR ultimately modulate the activation of p53 and the apoptotic pathway. When the level of damage is insufficient to activate apoptosis, its effects may still be observed as neuronal cell cycle activation [247]. Several studies have demonstrated that following different forms of stress such as, stroke, traumatic brain injury (TBI), MCI, and early AD [247,248,249,250,251,252], neurons may aberrantly re-enter the cell cycle at the G1 or S phases, causing permanent changes to neuronal metabolism and the development of a senescence-like state [247,253,254]. Once in the senescence-like state, these neurons release various signaling factors and toxins that damage neighboring cells, while instigating internal positive feedback loops of mitochondrial dysfunction and ROS production that reinforce persistent activation of the DDR [255,256,257]. In this way, the dysfunction of DSB-induced senescence-like cells and the paracrine effects of those cells on surrounding tissues may help explain how DNA damage can disrupt the delicate neural networks required for higher cognition. 

Alternatively, DSBs may affect the aging brain through the adult neural stem cell population. Adult neuronal stem cells (NSCs), also called radial glia-like cells (RGLs), are primarily located in the subventricular zone of the lateral ventricles and the subgranular zone of the dentate gyrus, where they are believed to contribute new interneurons to the striatum and dentate gyrus, respectively [258,259,260]. While the subventricular zone stem cells support the maintenance of the olfactory bulb, the subgranular zone of the hippocampus is critical for spatial learning, memory, and mood regulation [261,262,263,264]. Importantly, many studies have demonstrated that significant declines in the number of NSCs are closely associated with advancing age and neurodegenerative diseases [260,265,266]. Whereas neurons exist only in a post-mitotic state, NSCs can adopt several states, such as quiescent, activated, and differentiated. Several studies have demonstrated that advanced age disrupts the balance of these states, ultimately leading to diminished regenerative capacity [266,267]. DNA damage, including DSBs, is thought to contribute to the age-related demise of these NSCs by disrupting the natural stem cell state. In dividing cells, replication-induced stress is a major contributor to genomic instability and mutational burden [268]. NSCs typically avoid this stress by entering a quiescent state. Upon stress induction from tissue and DNA damage, however, adult NSCs are stimulated into cell cycle re-entry to replenish damaged cells, and in so doing, accumulate replication-induced mutations [269,270]. This mechanism is believed to explain the significant increase in genetic abnormalities found in NSCs isolated from aged mice when compared to those from younger mice [271].

Another explanation linking adult NSCs to age-related decline is based on observations that the efficiency of DNA repair mechanisms declines with age [266]. Because NSCs fail to proliferate and appropriately differentiate in the absence of functional DNA repair mechanisms [272], maintenance and deployment of adult NSCs are likewise perturbed in the aged brain [273]. The precise mechanisms underlying age-related declines in DNA repair pathways are unclear. Multiple studies have demonstrated evidence indicating transcriptional repression repair proteins, or age-related decreases in enzyme activity may contribute to pathway disruption [274,275], thus suggesting a possible role for currently unidentified accessory proteins such as those from the hnRNP family.

As hnRNP family proteins are involved in multiple fundamental cellular mechanisms, such as metabolism, inflammation, genome stability, identification of suitable aging-associated disease mechanism-oriented biomarkers, both physical and molecular, in susceptible individuals can provide clinicians with an early hint of pathological onset and buffer time to pre-determine the therapeutic strategy [276,277,278].

## 4. Targeted DNA Repair Therapeutics

Historically, DNA-targeted therapeutics have been relegated to the treatment of neoplastic pathologies. As new evidence has been reported and the link between genome maintenance and neurodegenerative disease has strengthened, it seems we are closer than ever to discovering potential disease-modifying therapies aimed at DNA damage and repair. Some contemporary approaches to DNA-based therapies for neurologic disease can be loosely categorized by target strategy: DNA damage-induced signaling, RNA metabolism, and chromatin modifications. Some of the earliest approaches to DNA repair-based therapies in neurologic disease targeted enzymes controlling chromatin organization. Histone deacetylase (HDAC) inhibitors comprise a major drug class but have relatively weak evidence of beneficial effects on neuronal genome stability. Some studies have demonstrated that increased acetylation of H4K16 is sufficient to disrupt 53BP1 recruitment to DSB sites while simultaneously increasing BRCA1 recruitment. Despite these changes, HDAC inhibition in neurons leading to H4K16 acetylation also decreases NHEJ efficiency and demonstrates increased γH2AX staining [279,280]. Nevertheless, the potential still exists for HDAC inhibition in neuronal DNA repair. As other studies have pointed out, HDAC2 inhibition is associated with improved learning and memory, increased number of synapses, and enhanced activity-induced transcription in mouse models of aging and neurodegeneration [279,281].

Targeting the DDR signaling network is perhaps the most well-developed area of DNA therapeutic ideas. Persistent activation of the DDR has clear negative effects on neuronal homeostasis but is also required for the repair of similarly deleterious mutations. Given the non-dividing state of post-mitotic neurons, it seems feasible that these cells may tolerate certain levels of DNA damage but are instead forced into more acutely harmful cell cycle re-entry or apoptosis secondary to DDR overactivation. To this end, targeted suppression of the DDR may prove beneficial for preserving neuronal function despite the presence of DNA damage. Early studies touched on this notion by identifying the neuroprotective role of caffeine, a nonspecific ATM kinase inhibitor, against etoposide-induced primary DSBs in cultured neurons [282]. Similarly, inhibiting the cyclin-dependent kinases (CDKs) involved in DNA damage-induced cell cycle re-entry also prevented neuronal death in a cerebral ischemia model [283]. As previously discussed, more recent studies have reported how the ablation of p53 in a mouse model of C9orf72 ALS/FTD completely reversed neurodegenerative changes and increased survival [193]. Attenuation of the DDR also appears beneficial for neuronal DSBs. Recent reports have demonstrated that blocking DSB recognition by the MRN complex not only confers a survival advantage to neurons but may also promote vital regenerative capacity. Specifically, the genetic knockdown or pharmacologic inhibition of MRE11 exonuclease by mirin, or the ATM kinase by KU-60019, were effective in mitigating neurodegenerative phenotypes in drosophila models. Furthermore, similar inhibition prevented the neuronal loss in DSB-containing primary hippocampal neuron cultures and decreased markers of apoptosis while stimulating axonal regeneration in rat models of the spinal cord and optic nerve injury [284]. As previously mentioned, the STING-IRF3 axis constitutes type-1 interferon-mediated inflammation that occurs in parallel with the ATM-NF-κB pathway. Recent studies have suggested that C9orf72 may be a therapeutic target in ALS/FTD with the effect of suppressing STING-mediated inflammation, particularly in the context of autoimmune activation [285,286]. Specifically, it was demonstrated that the cGAS-STING pathway activation is a critical step in promoting neuroinflammation via microglial recruitment in neurodegenerative and aging-associated conditions [287,288]. Additionally, it is known that *C9ORF72* repeat expansions induce TDP-43 proteinopathy [289], one effect of which is the release of mitochondrial DNA, which in turn activates the cGAS-STING system [139]. The resulting inflammation leads to a neuronal senescent-like phenotype and apoptosis [290]. In this way, it is conceivable that targeting C9orf72 may prove to be a useful therapeutic target for ALS/FTD.

Another DNA target involves DNA damage associated with RNA metabolism. As previously mentioned, the hnRNP family of proteins is closely associated with RNA processing at multiple levels, and some members are linked to specific RNA processing deficiencies. TDP-43, for example, has a role in ALS-associated microRNA dysregulation and DNA repair defects [291]. In this context, the fluroquinolone derivative enoxacin has been proposed to ameliorate neuromuscular dysfunction by facilitating microRNA processing [292]. A more recent study also highlights the potential of enoxacin to affect DSB repair via RNA processing mechanisms. Upon DSB induction, small non-coding RNAs, termed DNA damage response RNAs (DDRNAs), are generated in a DROSHER/DICER-dependent manner. Although these DDRNAs are not required for DSB recognition, there is strong evidence indicating their role in amplifying the DDR [293]. In IR-treated U2OS cells, enoxacin exposure enhanced the ATM-CHK2-P53 signaling axis, increased recruitment of DDR factors to break sites, and promoted NHEJ repair while decreasing the HR repair without altering the cell cycle state [294]. Subsequent sequencing analysis of the cell lines also indicated that the enoxacin exposure increased the accuracy of NHEJ-mediated repair compared to controls. Importantly, similar findings were observed in IR-treated primary cultures of mouse cortical neurons in addition to findings of decreased γH2AX staining. Taken together, these reports suggest that modulation of DDRNAs by enoxacin can rescue neuronal DNA damage by regulating the repair pathway choice which favors increased repair efficiency.

## Figures and Tables

**Figure 1 ijms-23-04653-f001:**
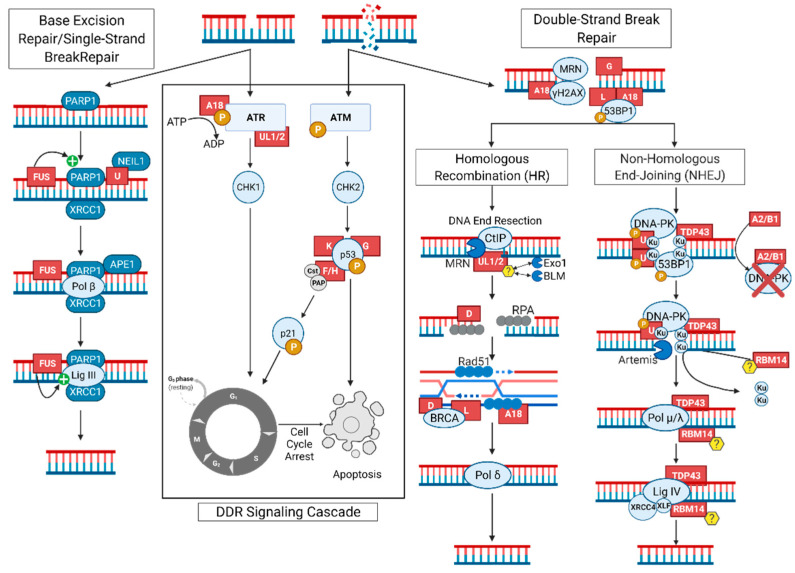
Summary of hnRNP Functions in DNA repair. Most hnRNP proteins exhibit significant functional overlap across the major SSB and DSB repair pathways in addition to damage-induced signaling cascades mediated by upstream ATM and ATR kinases. Specific functions of each protein are detailed in Table 1. MMEJ is an alternative DSB repair pathway important to neuron genome stability (not shown). hnRNP proteins are indicated by dark red text boxes and identified by their named letter (e.g., hnRNP-K is represented by “K”) or initialism. Other repair factors are presented in blue. Created with BioRender.com.

**Table 1 ijms-23-04653-t001:** Summary of hnRNP Roles in genome maintenance.

hnRNP Protein	Specific Role in the DDR and/or DSB Repair	Citations
TDP-43	Participates in DSB recognition by facilitating phosphorylation of H2AX Promotes recruitment of early NHEJ repair proteins 53BP1 and DNA-PKcs to DSB sites Permits DNA damage-induced nuclear translocation of NHEJ ligation complex and its specific recruitment to DSB sites Prevents transcription-dependent accumulation of R-loops and subsequent defects in DNA repair and replication in human dividing cell lines Associates with HDAC1 to maintain normal cell cycle activity and prevent DNA damage accumulation in a transgenic mouse model of FTLD	Konopka et al., 2020a Mitra et al., 2019 Guerrero et al., 2019 Wu, C. et al., 2020
P/P2/FUS/TLS	Participates in recognition of DSBs Facilitates efficient NHEJ and HR repair, possibly by recruiting HDAC1 to damage sites and enhancing its pro-DNA repair activity Enhances PARP-1 activity following oxidative DNA damage Recruits XRCC1/DNA Ligase3 to DNA damage sites and directly enhances ligation activity of DNA Ligase 3 Exerts transcription regulation of multiple DDR related proteins	Wang, H. et al., 2018 Wang. W. et al., 2013a Sukhanova et al., 2020 Mastrocola et al., 2013
hnRNP-U/SAF-A	Facilitates NHEJ repair by interacting with Ku70 in its phosphorylated form Facilitates BER repair by interacting with NEIL1 in its nonphosphorylated form Functions as a molecular switch ensuring NHEJ-mediated repair of DSBs occurs before BER repair in IR-treated cells Facilitates resolution of RNA:DNA hybrid structures (R-loops) formed during HR-mediated DSB repair	Hegde et al., 2016 Britton et al., 2014a
A1	Prevents aberrant DDR activation to telomeric structures by facilitating association of telomerase with 3′ telomeric ends, enhancing telomerase activation, and promoting the formation of the Shelterin complex Coordinates with proteins SRSF10 and Sam68 to alter transcript splicing of pro-apoptotic genes following DDR activation	Clarke et al., 2021 Ghosh and Singh, 2018 Sui, J. et al., 2015 Wang, T. et al., 2019
A2/B1	Negatively regulates DNA-PK activity following DSB induction Overexpression delays repair kinetics of IR-induced DNA damage	Iwanaga et al., 2005a Kamma et al., 2001 Liu & Shi, 2021
C/C1/C2	Maintains expression of HR repair-associated proteins BRCA1, BRCA2, RAD51, and BRIP, likely by preventing Alu exonization-induced nonsense-mediated decay of nascent mRNA transcripts Demonstrates RNA-dependent recruitment to DNA damage sites and interaction with HR-related PALB2/BRCA repair complexes. Cellular depletion causes diminished HR and enhanced alt-EJ-mediated DSB repair	Anantha et al., 2013
D/AUF1	Promotes HR-mediated DSB repair by facilitating DNA end resection processing of DSB ends Facilitates resolution of R-loop RNA:DNA hybrid structures, possibly in conjunction with hnRNP-U	Alfano et al., 2019a
F/H	Facilitates escape of p53 expression from DNA damage-induced global transcriptional repression by enhancing p53 pre-mRNA 3′-end processing and recruitment of CstF and PAP factors essential for cleavage and polyadenylation of p53 transcripts Indirectly affects the expression of DDR and apoptosis-related genes by enabling p53 expression	Decorsière et al., 2011
G/RBMX	Enhances DNA end joining repair fidelity in a p53 dependent manner by preventing nuclease degradation of ssDNA and dsDNA ends via direct binding. Indirectly promotes HR-mediated DSB repair via its RRM domain Enhances transcription of BRCA2	Shin et al., 2007 Adamson et al., 2012
K	Promotes transcriptional activation of p53 and p21 in an ATR-dependent manner	Lee, Seong Won et al., 2012 Pelisch et al., 2012
L	Enhances NHEJ and HR-mediated DSB repair, in part by promoting recruitment of 53BP1 and BRCA1 to DNA damage sites	Hu et al., 2019 Hu et al., 2015
UL1, UL2	Promotes ATR-dependent DDR signaling Promotes HR-mediated repair by enhancing DNA end resection, possibly by interaction with EXO1nuclease and BLM helicase	Gurunathan et al., 2015 Hong et al., 2013a Polo et al., 2012
CIRBP/A18	Modulates DDR signaling in response to cell stressors Promotes NHEJ and HR repair by enhancing recruitment of repair proteins to damaged sites in a PARP1-dependent manner	Chen et al., 2018 Lee, Hae Na et al., 2015 Sun et al., 2021 Yang, C. and Carrier, 2001 Yang, R. et al., 2010
RBM14	Promotes the NHEJ-mediated DSB repair by facilitating dissociation of Ku70/80 from DNA ends and/or the docking of DNA Ligase 4 complex Contributes to error-free NHEJ by facilitating recruitment and co-activation of RNAPII at damaged sites	Simon et al., 2017 Jang et al., 2020

## Data Availability

Not applicable.

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
