# Peer review of "DNA Double-Strand Breaks as Pathogenic Lesions in Neurological Disorders"

_ijms, 2022, doi:10.3390/ijms23094653_

Round 1

Reviewer 1 Report

In their review article, Provasek and colleagues update our knowledge on the complexities and peculiarities of DDR and genome repair mechanisms in the CNS and causally link the underlying molecular processes to the manifestation of degenerative CNS pathologies and disease propagating brain aging. By focusing on the CNS and neurons, they target the yet unmet need for a better understanding of genomic maintenance in a post-mitotic environment. With new dimensions, the authors highlight the role of RDBPs and hnRNP family members in DDR signaling and genome protection. Finally, they discuss DDR and DNA repair in the frame of putative new therapeutic options.

This is an excellent, highly informative review article, which complements and expands contemporary literature on the importance of genome maintenance in CNS function and disease. Overall, this comprehensive review article is well-structured, embeds ample literature, and connects detailed target knowledge with clinical disease phenotypes.

Prior to publication, I recommend consideration of the following aspects:

Major:

- Abstract, line 17: According to standard concepts, HR I still held exceptional to serve as DSB repair mechanism in differentiated neurons. This should be clarified here.

- End of introduction: Advantageous would be a short preview on the following content. Currently, it is not clear where the introduction ceases.

- Passage 2.2.: Suggested is to start with a brief summary on the nomenclature of hnRNP family members (e.g., according to Geuens et al., Hum Genet, 2016).

- Line 720 onwards: As differentiated neurons are primarily dependent on NHEJ, how can BRCA1- and HR- linked repair deficiencies explain neuronal dysfunction and loss in AD pathology? Results from N2a murine neuroblast and their endowment with repair cascades might not suffice to explain the situation in mature cortical brain neurons. If restricted to hippocampal stem cell niches, this should be indicated.

- Line 510; line 840 onwards: It is still a matter of research whether senescence criteria including secretory SASP profiles in postmitotic neurons are identical to the cellular and molecular phenotypes characteristic of classical senescence, coined in replication-competent cell entities. Thus, I suggest a terminology restricted to ‘a senescence-like phenotype’ in neurons. The topic of postmitotic cellular senescence (PoMiCS) is currently under detailed investigation, thus merits broader citations (e.g., Jurk et al., Aging Cell 2012; Anderson et al., EMBO, 2019; Sapieha, Mallette, Trends Cell Biol. 2018; Von Zglinicki et al., Antioxid. Redox Signal. 2021; Wengerodt et al., Cells 2019).

- Line 818 onwards, discussion: The authors correctly state that WHO declared aging as a cause of disease. Beyond, there is the scientific call to raise aging to a disease level. Though appearing preterm in face of the lack of a universal definition of aging, which has not been anchored in the WHO ICD11 (set into effect 01/2022), increasing knowledge on DDR and DNA damage might help establish a biomarker consensus for aging. Such ‘diagnostic’ addition might complement the authors’ preview on therapeutics discussed in their review.

- Discussion: I suggest the implementation of current knowledge on extrachromosomal circular DNA, a byproduct of mutational events and DNA repair processes, which is also linked to the DNA sensing cGas/STING response. Prognostic and therapeutic impact of this DNA species is well described in tumors. Considering the spectrum of origin also encompassing transcriptional activity apart from replication stress, it appears appealing to discuss such novel DNA entity in the context of CNS disorders (see, e.g., Shibata et al., Science, 2012; Dillon et al., Cell, 2015; Kim et al., Nat Gen 2020; Ain et al., IJMS, 2020; Wang et al., Nature, 2021).

- In the introduction, the authors mention the possibility of homology-directed repair processes in post-mitotic neurons. Moreover, transcription-coupled DSB repair in genes causative in HD are discussed (line 69; line 245). Suggested in this context is to cite additional work, where a transcription-coupled, homology-directed but replication-independent repair process in G0/G1 is described for neurons, involving RPA1, Rad51/52 and CSB proteins (Welty et al., J Biol Chem, 2018; Wei et al., PNAS, 2015). Such process might give further insight into error-free repair choices in postmitotic neurons.

- The passage on c-GAS/STING (line 512 onwards) would profit from more directly delineating the link to the RDBP TDP-43 (ref.131) and the association with C9ORF72 pathology (McCauley et al., Nature, 2020). In a broader scope, its crucial immune response is consolidated for diverse neurodegenerative CNS disorders including ALS/FTD and linked to SASP in senescence, for which it is discussed as a future therapeutic target (Decout et al., Nat Rev., 2021; Fryer et al., Front Neurosci, 2021).

Minor:

- Figure 1: Needs connection to the text by referencing at appropriate position. For detailed comprehension, it would profit from format enlargement. The legend requires explanation as for majuscules indicating the diverse hnRNP family members.

- Table 1: Width of horizontal columns might be optimized according to the text features; citations without brackets.

- Abstract, line 17, etc.: ‘micro-homology’ consistently with hyphen (there are several versions: ‘microhomology’, ‘micro homology’, ‘micro-homology’)

- Abstract, line 20: full name is ‘heterogeneous nuclear ribonucleoproteins’

- Abstract, line 23: ‘selected’ instead of ‘select’

- Introduction, line 77: Pol-q encodes Pol-θ, please adjust

- Introduction, line 83: delete surplus space

- Introduction, line 94: place-holder ‘ref.’ needs a citation index

- Line 208: ‘BRCT’ and ‘MDC1’ need explanation prior to abbreviation; please check throughout

- Caption, line 255: for consistence: ‘RNA/DNA binding proteins’ as later referred in the text

- Line 246: sentence is too long, set a dot after ‘pathology’ before ‘RIF1’

- Line 369: close the sentence and set a dot after ‘BRIP’.

- Line 379, line 409: ref. 96-98 and 104-107 in brackets are doubled, delete one of each

- Line 439: substitute ‘mu’ by greek ‘µ’. Please check nomenclature as for appropriate use of Pol, pol, or POL

- Line 459-62: sentence is cropped, thus needs revision

- Line 486: NHEJ is error-prone, please correct

- Line 550: typo, change ‘disceet’ to ‘discrete’

- Lines 559, 622, 784: delete surplus commas after ‘expected’ and before ‘which’

- Line 598: more precise: first intron/promoter region; genes in italics throughout

- Line 657: ‘etc.’ should be avoided; replace, e.g., by ‘and others’, or complete by ‘apraxia’, ‘aphasia’ (according to ICD 10 criteria)

- Line 700: ‘mitophagy’ with minuscule

- Line 710: delete space after brackets

- Line 773: BDPs instead of ‘BPDs’

- Line 841: delete apostrophe, change DSB’s to DSBs

- Line 903-904: add tab between ‘pair’ and ‘based’

- Nomenclature: according to FUS and TDP-43 etc, ‘C9orf72’ with majuscules, in case it is discussed in the human context; genes should generally appear in italics

Author Response

This is an excellent, highly informative review article, which complements and expands contemporary literature on the importance of genome maintenance in CNS function and disease. Overall, this comprehensive review article is well-structured, embeds ample literature, and connects detailed target knowledge with clinical disease phenotypes. Prior to publication, I recommend consideration of the following aspects:

Response: We sincerely thank the reviewer for the appreciation and for the critical comments and suggestions. We have carefully revised the manuscript by accommodating all the suggestions. These comments have significantly helped us to improve the review and we express gratitude for the reviewer’s time and insights.

Major:

- Abstract, line 17: According to standard concepts, HR I still held exceptional to serve as DSB repair mechanism in differentiated neurons. This should be clarified here.

Response: Thanks for pointing out this. We have modified the statement in the revised manuscript,

Mammalian cells engage with several different strategies to recognize and repair chromosomal DSBs based on the cellular context and cell cycle phase, including homologous recombination (HR)/homology-directed repair (HDR), microhomology-mediated end-joining (MMEJ), and the classic non-homologous end-joining (NHEJ).

- End of introduction: Advantageous would be a short preview on the following content. Currently, it is not clear where the introduction ceases.

Response: Thanks for pointing it out. We have now expounded on the ending of the Introduction by mentioning some of the salient topics covered in Section 1 “In this section, we will describe how neurons in basal and pathological states respond to breaks in both strands of the DNA molecule, and how these breaks affect brain physiology. Furthermore, we will highlight the roles of key proteins involved in the recognition, repair, and signaling of neuronal DSBs, as well as discuss how these processes are affected by chromatin dynamics.

- Passage 2.2.: Suggested to start with a brief summary on the nomenclature of hnRNP family members (e.g., according to Geuens et al., Hum Genet, 2016).

Response: Thanks for your critical suggestion. We have addressed this inclusion in lines 268-274.

The hnRNP family was founded on the discovery of hnRNPs A/B and C, with later studies identifying a further 18 major protein members. Subsequent additions to the hnRNP family have generally been denoted by a letter to indicate the order of discovery or structural similarity to previously identified members; this naming convention is not universal, however, as other members such as TDP-43 and FUS lack both the hnRNP prefix and letter identifier [65,66]. While hnRNPs are classically recognized for their RNA binding activities, they have also been proven vital to successful DNA repair.

- Line 720 onwards: As differentiated neurons are primarily dependent on NHEJ, how can BRCA1- and HR- linked repair deficiencies explain neuronal dysfunction and loss in AD pathology? Results from N2a murine neuroblast and their endowment with repair cascades might not suffice to explain the situation in mature cortical brain neurons. If restricted to hippocampal stem cell niches, this should be indicated.

Response: We apologize for not clearly mentioning this point. We have added further information on BRCA1 and brain development/neuronal plasticity in lines 756-764, stating that BRCA1-related DSB repair defect is a primary factor for cell cycle proficient neurons, like hippocampal neurons and neural progenitor cells, because of its essential connection with homologous recombination repair pathway.

Notably, the tumor suppressor breast cancer susceptibility gene 1 (BRCA1)-ATM kinase signaling axis plays crucial roles in the development of brain function and size by modulating the polarization of neural progenitor cells (NPC) [193]. Given the essential role of BRCA1 in the HR-mediated DSB repair [194] and exploitation of this pathway by NPCs toward neuronal plasticity related to cognitive functions [195],  BRCA1 pathology seems more relevant to the hippocampal and entorhinal cortex neurons as well as migratory NPCs from the subventricular zone in the AD and dementia. Further research is needed to understand whether this mechanism can affect the motor neurons, also, and if yes, then to what extent.

Line 510; line 840 onwards: It is still a matter of research whether senescence criteria including secretory SASP profiles in postmitotic neurons are identical to the cellular and molecular phenotypes characteristic of classical senescence, coined in replication-competent cell entities. Thus, I suggest a terminology restricted to ‘a senescence-like phenotype’ in neurons. The topic of postmitotic cellular senescence (PoMiCS) is currently under detailed investigation, thus merits broader citations (e.g., Jurk et al., Aging Cell 2012; Anderson et al., EMBO, 2019; Sapieha, Mallette, Trends Cell Biol. 2018; Von Zglinicki et al., Antioxid. Redox Signal. 2021; Wengerodt et al., Cells 2019).

Response: We thank the reviewer for mentioning this critical point. As suggested, we have modified the section 3.3 by adding the following sentences:

Given that telomere shortening as a common mechanism for cellular senescence, it has been shown that post-mitotic cells/tissues may undergo a telomere length-independent damage, leading to the actual senescence stage via a non-canonical senescence-linked secretory pro-hypertrophic and fibrotic phenotype [220,221]. It remains an open-ended question how the senescence mechanism in the post-mitotic cells can be accelerated by the biological aging processes [222].

-Line 818 onwards, discussion: The authors correctly state that WHO declared aging as a cause of disease. Beyond, there is the scientific call to raise aging to a disease level. Though appearing preterm in face of the lack of a universal definition of aging, which has not been anchored in the WHO ICD11 (set into effect 01/2022), increasing knowledge on DDR and DNA damage might help establish a biomarker consensus for aging. Such ‘diagnostic’ addition might complement the authors’ preview on therapeutics discussed in their review.

Response: We thank the reviewer for mentioning this important point. As suggested, we have added the following sentences in lines 954-958;

Since hnRNP family proteins are involved in multiple fundamental cellular mechanisms like metabolism, inflammation, genome stability, identification of suitable aging-associated disease mechanism-oriented biomarkers, both physical and molecular, in the susceptible individuals can provide clinicians with early hint of pathological onset and a buffer time to pre-determine the therapeutic strategy [265-267]

- Discussion: I suggest the implementation of current knowledge on extrachromosomal circular DNA, a byproduct of mutational events and DNA repair processes, which is also linked to the DNA sensing cGas/STING response. Prognostic and therapeutic impact of this DNA species is well described in tumors. Considering the spectrum of origin also encompassing transcriptional activity apart from replication stress, it appears appealing to discuss such novel DNA entity in the context of CNS disorders (see, e.g., Shibata et al., Science, 2012; Dillon et al., Cell, 2015; Kim et al., Nat Gen 2020; Ain et al., IJMS, 2020; Wang et al., Nature, 2021).

Response: We thank the reviewer for this valuable suggestion. We have modified accordingly in the revised manuscript in lines 559-573.

Furthermore, initially discovered in both the normal and cancer cell lines, extrachromosomal circular DNA (eccDNA) has been identified to cross-talk with STING-associated pro-inflammatory pathways [136-138]. The eccDNA is found to derive from the transcriptionally active, exon rich and non-repetitive DNA sequences [139]. Interestingly, eccDNA production does not involve neither NHEJ nor HR machinery, rather mismatch repair factor MSH3 has been implicated in regulating the cellular load of eccDNA molecules [139]. Although emerging studies have indicated that eccDNA may play multi-faceted regulatory roles in the cell, including apoptosis [137], and may have mechanistic linkage with aging and neurodegenerative diseases, however, the precise role of hnRNP family proteins and their pathological mutations in eccDNA production needs to be investigated in detail in the future [140]. As studies reveal, its origin in the post-mitotic neurons might be related to the R-loop dysregulation in the transcriptionally active genomic regions [142]. However, in another study by Zhu et al. [143], eccDNA is found to produce from DNA DSBs flanking short microhomology sequence, suggesting critical role of NHEJ in combination with MMEJ, especially in the post-mitotic neurons.

- In the introduction, the authors mention the possibility of homology-directed repair processes in post-mitotic neurons. Moreover, transcription-coupled DSB repair in genes causative in HD are discussed (line 69; line 245). Suggested in this context is to cite additional work, where a transcription-coupled, homology-directed but replication-independent repair process in G0/Gis described for neurons, involving RPA1, Rad51/52 and CSB proteins (Welty et al., J Biol Chem, 2018; Wei et al., PNAS, 2015). Such process might give further insight into error-free repair choices in postmitotic neurons.

Response: We thank the reviewer for this important suggestion. We have modified accordingly.

- The passage on c-GAS/STING (line 512 onwards) would profit from more directly delineating the link to the RDBP TDP-43 (ref.131) and the association with C9ORF72 pathology (McCauley et al., Nature, 2020). In a broader scope, its crucial immune response is consolidated for diverse neurodegenerative CNS disorders including ALS/FTD and linked to SASP in senescence, for which it is discussed as a future therapeutic target (Decout et al., Nat Rev., 2021; Fryer et al., Front Neurosci, 2021).

Response: We thank the reviewer for this important suggestion. We have modified accordingly in the revised manuscript, in lines 1021-1031, as follows:

As mentioned earlier that the STING-IRF3 axis constitutes the type-1 interferon-mediated inflammation, in parallel to the ATM-NF-κB pathway, recent studies have shown that C9orf72 could be a therapeutic target in ALS/FTD to suppress the STING-mediated immune-inflammation, particularly during the autoimmune activations [282,283]. Emerging studies have shown that the cGAS-STING pathway activation is a critical step in promoting neuroinflammation via microglial involvement in neurodegenerative and aging-associated pathological conditions [284,285]. Since C9ORF72 repeat expansion induces TDP-43 proteinopathy [286], which in turn activates the cGAS-STING system by releasing mitochondrial DNA in the affected motor neurons [287], resulting in the neuronal senescence and apoptosis [288],  targeting C9orf72 could be a potential therapeutic aspect in ALS/FTD and related dementia.

Minor:

- Figure 1: Needs connection to the text by referencing at appropriate position. For detailed comprehension, it would profit from format enlargement. The legend requires explanation as for majuscules indicating the diverse hnRNP family members.

Response: Modified as suggested.

- Table 1: Width of horizontal columns might be optimized according to the text features; citations without brackets.

Response: Modified as suggested.

- Abstract, line 17, etc.: ‘micro-homology’ consistently with hyphen (there are several versions: ‘microhomology’, ‘micro homology’, ‘micro-homology’)

Response: Modified as suggested.

- Abstract, line 20: full name is ‘heterogeneous nuclear ribonucleoproteins’

Response: Modified as suggested.

- Abstract, line 23: ‘selected’ instead of ‘select’

Response: Modified as suggested.

- Introduction, line 77: Pol-q encodes Pol-θ, please adjust

Response: Modified as suggested.

- Introduction, line 83: delete surplus space

Response: Modified as suggested.

- Introduction, line 94: place-holder ‘ref.’ needs a citation index

Response: Modified as suggested.

- Line 208: ‘BRCT’ and ‘MDC1’ need explanation prior to abbreviation; please check throughout

Response: Modified as suggested.

- Caption, line 255: for consistence: ‘RNA/DNA binding proteins’ as later referred in the text

Response: Modified as suggested.

- Line 246: sentence is too long, set a dot after ‘pathology’ before ‘RIF1’

Response: Modified as suggested.

- Line 369: close the sentence and set a dot after ‘BRIP’.

Response: Modified as suggested.

- Line 379, line 409: ref. 96-98 and 104-107 in brackets are doubled, delete one of each

Response: Modified as suggested.

- Line 439: substitute ‘mu’ by greek ‘µ’. Please check nomenclature as for appropriate use of Pol, pol, or POL

Response: Modified as suggested.

- Line 459-62: sentence is cropped, thus needs revision

Response: Modified as suggested.

- Line 486: NHEJ is error-prone, please correct

Response: Modified as suggested.

- L Response: Modified as suggested.

ine 550: typo, change ‘disceet’ to ‘discrete’

- Lines 559, 622, 784: delete surplus commas after ‘expected’ and before ‘which’

Response: Modified as suggested.

- Line 598: more precise: first intron/promoter region; genes in italics throughout

Response: Modified as suggested.

- Line 657: ‘etc.’ should be avoided; replace, e.g., by ‘and others’, or complete by ‘apraxia’, ‘aphasia’ (according to ICD 10 criteria)

Response: Modified as suggested.

- Line 700: ‘mitophagy’ with minuscule

Response: Modified as suggested.

- Line 710: delete space after brackets

Response: Modified as suggested.

- Line 773: BDPs instead of ‘BPDs’

Response: Modified as suggested.

- Line 841: delete apostrophe, change DSB’s to DSBs

Response: Modified as suggested.

- Line 903-904: add tab between ‘pair’ and ‘based’

Response: Modified as suggested.

- Nomenclature: according to FUS and TDP-43 etc, ‘C9orf72’ with majuscules, in case it is discussed in the human context; genes should generally appear in italics

Response: Modified as suggested.

Reviewer 2 Report

This is a truly outstanding and comprehensive review on recombination in neurological disorders.  The authors do an outstanding job in defining the mechanisms for HR and NHEJ.  I wish that there was more discussions related to Alt-NHEJ.  However, this is a minor point.  In addition, the authors provide excellent and comprehensive discussions on neurological diseases such as Alzheimer's disease and ischemia.  Again, I wish that the authors could discussion recombination within the context of neurological cancers such as astrocytomas.  However, adding that information may be a bit excessive.  

Author Response

We thank the reviewer for the appreciation and for recommendation. We have included few additional details on DSB repair pathways and their regulation in general including Alt-EJ.